# Lipid Nanocarriers for Anti-HIV Therapeutics: A Focus on Physicochemical Properties and Biotechnological Advances

**DOI:** 10.3390/pharmaceutics13081294

**Published:** 2021-08-19

**Authors:** Maria J. Faria, Carla M. Lopes, José das Neves, Marlene Lúcio

**Affiliations:** 1CF-UM-UP, Centro de Física das Universidades do Minho e Porto, Departamento de Física da Universidade do Minho, 4710-057 Braga, Portugal; faria.mariajf@gmail.com; 2FP-I3ID, FP-ENAS/CEBIMED, Fernando Pessoa Energy, Environment, and Health Research Unit/Biomedical Research Center, Portugal and Faculty of Health Sciences, Fernando Pessoa University, 4200-150 Porto, Portugal; 3i3S, Instituto de Investigação e Inovação em Saúde, Universidade do Porto, 4200-135 Porto, Portugal; j.dasneves@i3s.up.pt; 4INEB, Instituto de Engenharia Biomédica, Universidade do Porto, 4200-135 Porto, Portugal; 5CESPU, Instituto de Investigação e Formação Avançada em Ciências e Tecnologias da Saúde, 4585-116 Gandra, Portugal; 6CBMA, Centro de Biologia Molecular e Ambiental, Departamento de Biologia, Universidade do Minho, 4710-057 Braga, Portugal

**Keywords:** ARV delivery, biotechnology in ARV, biological barriers, lipid emulsions, lipid nanoparticles, liposomes, RNAi and ARV codelivery

## Abstract

Since HIV was first identified, and in a relatively short period of time, AIDS has become one of the most devastating infectious diseases of the 21st century. Classical antiretroviral therapies were a major step forward in disease treatment options, significantly improving the survival rates of HIV-infected individuals. Even though these therapies have greatly improved HIV clinical outcomes, antiretrovirals (ARV) feature biopharmaceutic and pharmacokinetic problems such as poor aqueous solubility, short half-life, and poor penetration into HIV reservoir sites, which contribute to the suboptimal efficacy of these regimens. To overcome some of these issues, novel nanotechnology-based strategies for ARV delivery towards HIV viral reservoirs have been proposed. The current review is focused on the benefits of using lipid-based nanocarriers for tuning the physicochemical properties of ARV to overcome biological barriers upon administration. Furthermore, a correlation between these properties and the potential therapeutic outcomes has been established. Biotechnological advancements using lipid nanocarriers for RNA interference (RNAi) delivery for the treatment of HIV infections were also discussed.

## 1. Introduction

The human immunodeficiency virus (HIV) is known to promote the continuous deterioration of the host immune system, being responsible for the acquired immunodeficiency syndrome (AIDS) [1,2]. According to the Joint United Nations Program on HIV infection/AIDS (UNAIDS), by the end of 2018, the epidemic accounted for more than 30 million deaths worldwide with a particular incidence in the female population and Sub-Saharan African countries [3]. Currently, 37.9 million people are infected with the virus and only a fraction (≈82%) have access to antiretroviral (ARV) therapy [3].

ARVs revolutionized HIV infection/AIDS clinical history and their approval for therapeutic purposes transformed this condition into a chronically manageable disease [4]. ARV-based therapies continue to be the best treatment option against HIV infection/AIDS providing prolonged viral suppression and, consequently, lower mortality rates [5,6]. The first treatments were based on monotherapy regimens which rapidly led to the development of ARV resistance [7,8]. Consequently, novel strategies were adopted, namely, combined antiretroviral therapy (cART; formerly referred to as highly effective antiretroviral therapy (HAART)) based on the simultaneous administration of three or more different classes of drugs [7,8]. 

Although HIV infection/AIDS stands as a public health concern, anti-HIV therapies have greatly increased the life quality and expectancy of infected individuals [9]. However, these therapies are challenging and often difficult to implement in the developing world. Multiple factors may compromise their success, such as: (i) adverse effects associated with the multi-regimen therapies extended over long periods; (ii) development of viral resistance; (iii) ineffective viral suppression due to low drug concentrations in viral reservoirs; (iv) pharmacokinetic problems and possible interactions between drugs; (v) poor stability and reduced shelf-life; (vi) low patient adherence; (vii) unbearable high costs for most of the populations in need, and (viii) socio-cultural constraints that limit the access to these treatments [10,11]. 

In the last decades, several strategies to improve HIV disease management using nanotechnology have emerged and seen tremendous growth both in treatment and prevention. Nanotechnology-based systems radically changed the global medical scheme and gained considerable attention in therapeutic research. In particular, nanomedicine-based approaches may help to improve pharmacokinetic problems (e.g., low oral bioavailability or short half-life) of ARV drugs [9,12,13,14,15,16,17,18,19,20,21]. Poor aqueous solubility is another common problem transversal to many drugs, which can be improved by encapsulation in drug carriers [22]. The reduction of the particles size to the nanometric scale increases the surface area available for solvation which has shown to be an effective strategy to increase drug solubility and, consequently, improve oral bioavailability [23]. Another interesting feature brought by nanomedicine is the ability to modulate the drug release profiles to occur over a longer time and at higher effective doses to the specific sites [8,22]. Moreover, toxicity associated with ARV therapies may also be circumvented using drug-loaded systems. A possible explanation is the controlled release profiles obtained with nanocarriers, reducing the toxicity namely at the cellular level [2]. 

The encapsulation of ARV drugs is particularly interesting as a targeting strategy towards cellular and anatomic HIV reservoirs and it can be achieved either by passive or active targeting [2,8]. In the first case, the targeting is dependent on nanocarriers’ intrinsic properties such as mean diameter, surface properties (e.g., charge), and shape [2]. On the other hand, active targeting typically depends on the functionalization of the carrier surface with ligands that recognize receptors at the targeted tissues [2,9]. Furthermore, these carriers act as protective shields against external threats (e.g., chemical and enzymatic degradation) leading to increasing residence periods of ARV in the organism [9]. This feature may promote the reduction of required doses and, consequently, prolong time intervals between administrations [2]. Ultimately, it is possible to encapsulate different types of therapeutic payloads within the same system which can contribute to simpler administrations increasing patient adherence but also reduce possible errors related to therapeutic regimens [2,24].

Among the multiple nanocarriers that can be used for ARV delivery, lipid-based nanocarriers hold great promise since 15 of the 21 marketed approved nanomedicines are liposomes or lipid nanoparticles (AmBisome^®^, DaunoXome^®^, DepoCyt^®^, DepoDur^®^, Doxil^®^, Inflexal^®^ V, Marqibo^®^, Mepact^®^, Myocet^®^, Visudyne^®^, Abelcet^®^, Amphotec^®^, Fungizone^®^, Diprivan^®^, Estrasorb^®^) [25]. Of particular notice, this list has been recently upgraded with the introduction in the market of nucleoside-based nanomedicines for the treatment of hereditary transthyretin-mediated amyloidosis (Onpattro^®^) and prophylaxis of severe acute respiratory syndrome coronavirus 2 (SARS-CoV-2) infection (Pfizer/BioNTech Comirnaty^®^ and the Moderna COVID-19 vaccines). Furthermore, these carriers are well-accepted in the scientific community for therapeutic purposes mainly because their structural units are generally recognized as safe (GRAS) [25]. Additionally, lipids’ biocompatibility and biodegradability properties as well as their versatility make them suitable and safe delivery systems for humans, with low or non-associated toxicity [11,26]. A large number of lipid-based nanocarriers developed for ARV delivery justifies a constantly updated review. Even though some reviews have covered this topic [27,28,29], it is important to address some neglected aspects regarding the details of formulation development to serve as a guide for researchers working in this field. To the best of our knowledge, no similar reviews have considered the composition and characterization of lipid nanocarriers in terms of size, colloidal stability, encapsulation, and drug loading efficiency, as well as establishing a correlation between the nanocarriers’ physicochemical properties and their potential anti-HIV therapeutic outcomes. Moreover, biotechnological applications of lipid nanocarriers loaded with anti-HIV therapeutics will be presented including the use of lipoplexes for small interference ribonucleic acid (siRNA) delivery and other interesting prospects for other disease conditions (e.g., neurodegenerative diseases) that have not yet been considered. For example, considering that reverse transcriptase (RT) is found in a variety of human cells, including those in the brain, and that it is involved in somatic gene recombination (SGR), which is linked to dysregulated neuronal genomes in Alzheimer’s disease (AD), the inhibition of this enzyme by ARV agents in combination with siRNA-mediated silencing of its expression is considering a promising biotechnological approach for the prevention and/or treatment of this neurodegenerative disease. The utilization of lipid-based nanocarriers for co-delivery ARV and siRNA aids to cross the brain-blood barrier (BBB).

## 2. ARV Agents: Mechanism of Action and Limitations

Following the isolation and subsequent identification of HIV as the main agent responsible for the onset of HIV infection/AIDS, significant progress was made, allowing for a detailed characterization of the virus and its life cycle, as well as a better understanding of the mechanisms underlying its mode of action [30]. In this way, it became possible to identify new, highly specific pharmacological targets in the HIV life cycle (Figure 1) that allowed the development of the first drugs that would change the course of HIV infection/AIDS history.

In 1987, zidovudine (AZT) was approved as the first ARV for therapeutic use. Since then, and in a short period, 6 more classes of ARVs have been developed and, to date, 49 medicines containing single ARV drugs or drug associations (as in the case of cART) have been approved and made available on the market by the United States Food and Drug Administration (U.S. FDA) for the clinical treatment of HIV infection/AIDS [31]. The existing ARV drugs can be classified according to their target [4] (Figure 2, Figure 3, Figure 4, Figure 5 and Figure 6).

Thus, ARV can be grouped into the following therapeutic classes (Table 1): cell entry inhibitors (stage 1 of Figure 1 and Figure 3; fusion inhibitors (FI) (stage 2 of Figure 1 and Figure 3; nucleoside reverse transcriptase inhibitors (NRTI), non-nucleoside RT inhibitors (NNRTI) (stage 4 of Figure 1 and Figure 4); integrase inhibitors (IIs) (stage 5 of Figure 1 and Figure 5); and protease inhibitors (PI) (stage 6 of Figure 1 and Figure 6) [31,32]. Pharmacokinetic enhancing drugs (e.g., cobicistat) can also be used in association with ART agents to improve therapeutic effectiveness. 

Despite the recognized overall success of cART, particularly in developing countries, this therapeutic strategy continues to raise some important issues, and its effectiveness is affected by several limitations. For example, the multi-dose treatments administered over extended periods can result in the development of ARV resistance mechanisms and can also lead to the inability to effectively suppress the virus due to the difficulties in maintaining consistent drug levels, particularly in viral reservoirs [8,35]. The mechanisms underlying ARV resistance are often related to HIV genetic variability and HIV reverse transcriptase processing errors. This high mutation rate coupled with the virus’s fast replication leads to the creation of innumerable virus variants (quasispecies) capable of avoiding the immune system [36,37]. Additionally, the recombination of more than one viral strain during infection or the accumulation of proviral variants also contribute to viral resistance [37,38]. Although some HIV variants display primary mutations that make them less susceptible to ARV action, most ARV resistance results from direct exposure to these regimens and it was already observed in all six therapeutic classes through different mechanisms [36]. For example, in NRTI, whose main function is to block the viral RT and inhibit HIV replication, resistance can occur by two mechanisms. The first mechanism corresponds to mutations at or near the drug-binding site of RT (e.g., M184V, L74V, K65R, and others) leading to a conformational change in the enzyme that ultimately blocks the binding of NRTI to the active site [36,39]. Such a mechanism enables viral RT to discriminate between dideoxy-NRTI chain terminators and endogenous triphosphate deoxynucleosides, preventing the binding of NRTI to viral DNA [37] The second mechanism corresponds to phosphorylytic removal of NRTI-triphosphate from its site of attachment in the viral DNA chain [36,37]. These mutations are characteristic of the thymidine analogs (AZT and d4T) and can also be described as thymidine analog mutations (TAM). TAM can be further divided into type I (e.g., M41L, L210W, and T215Y) and II (e.g., D67N, K70R, T215F, and K219Q/E), however, type I is responsible for higher levels of phenotypic and clinical cross-resistance [37].

The most frequent mutations that occur in NNRTIs, take place within their binding pocket and mostly affect hydrophobic residues of HIV-RT [36,40]. These mutations (e.g., L100I, G190S/A, and Y181C) alter the binding site of RT to NNRTIs which consequently decreases the binding affinity and alters the overall structure of the complex [39]. Other mutations (e.g., K103N) can act using a different mechanism such as the establishment of a hydrogen bond at the entrance of the binding pocket. This helps to maintain the pocket closed and limits NNRTI diffusion. Nevertheless, one of the biggest hurdles related to the use of NNRTI is that the binding site in RT is approximately the same for all of them, which means that a single mutation can lead to high-level drug resistance and cause cross-resistance among all NNRTI [36,41].

Furthermore, the resistance to PI is associated with mutations in the protease gene and subsequent replacement of amino acids within the protease enzyme (e.g., D30N, V32I, G48V, V82A, and others) [36,41]. These modifications will reduce the binding affinity between the catalytic binding site of the enzyme and the PI [36]. Other mutations in the enzyme flap (e.g., I54M/L) and core (e.g., L76V and N88S) can also decrease PI susceptibility [40]. In response to these mutations, the geometry of the catalytic site of the homodimer is enlarged disabling the inhibitor to effectively bind to the gene and block cleavage [36]. 

The development of resistance in FI is related to mutations in gp41 codons 36–45 (e.g., G36DEV, V38EA, Q40H, N42T, and N43D), correspondent to the location where T-20 will bind [36,38]. In the case of II, the occurrence of mutations (e.g., N155H, Q148R, Y143R, and others) at Asp64 and Asp116 carboxylate residues (which coordinate an Mg^2+^ ion) compromise the catalytic activity of the enzyme [41]. It is thought that the functional group of IIs binds selectively to the enzyme complexes which further interferes with strand transfer of viral and host DNA [41]. Finally, cell entry inhibitors such as maraviroc (CCR5 inhibitor) may develop resistance via gp120 mutations, enabling HIV to bind the CCR5-CCR5 inhibitor complex [40]. However, the most common mechanism of resistance to CCR5 inhibitors is associated with an enhancement of CXCR4 tropic viruses that are intrinsically insensitive to CCR5 inhibitors [40].

Besides the resistance mechanisms, prolonged treatment regimens often result in poor adherence and careless patient intake, as well as treatments with high associated costs [2,42,43]. In addition to this, any interruption in the therapeutic regimen results in treatment failure and viral resistance. Moreover, drug associations can improve the therapeutic effectiveness, but they may also have the opposite effect due to an increase in deleterious drug-drug interactions. Prolonged use of ARV therapeutic regimens is also often associated with toxic side effects (e.g., constipation/diarrhea, abdominal pain, nausea, liver and metabolic disorders, kidney stones, anemia, fatigue, headache, fever, muscular dystrophy, and peripheral neuropathy) that compromise the quality of life of patients [44]. 

Pharmacokinetic issues are another significant limitation of cART and single ARV therapies. In either case, ARV demonstrates low and unpredictable bioavailability after oral administration due to their poor gastrointestinal (GI) absorption, extensive first-pass metabolism, and GI enzymatic degradation. The majority of ARV drugs are classified in biopharmaceutical class system (BCS) II, III, or IV, which means they have low solubility and permeability. For drugs given orally, solubility is required to confirm drug absorption and clinical response. The speed and extent of oral drug diffusion through the mucus layer, submucosa, and epithelial cell barriers into the blood or lymphatic circulation is referred to as permeability. Low solubility and permeability thereby show that ARV drugs are poorly absorbed in the body [45]. Even after absorption, most ARV present other significant pharmacokinetic limitations, such as a short half-life that demands recurrent dose administration in a fastidious dosage regimen, which contributes to poor patient adherence [2,44,45]. Another pharmacokinetic issue is poor body distribution of ARV, which prevents reaching certain target tissues that serve as viral reservoirs. High plasma protein binding of ARV, for example, impairs drug permeation across the BBB [46]. The central nervous system (CNS) is known as an anatomical viral reservoir, where HIV survives in long-lived cells, such as microglia. As a result, viral eradication by ARV drugs or cART becomes more difficult and newer and drug-resistant HIV strains develop. Furthermore, some ARV drugs’ inability to enter the CNS further restricts eradication. ARV drugs may also be expelled from CNS at the BBB level by efflux transporters like glycoprotein P (P-gp) [47]. Simultaneously, the inflammatory response induced by HIV infection of the CNS causes permeability increase of the BBB and plays an important role in neuropathogenesis [47,48]. 

Moreover, ARV fails to target lymphatic system cells (e.g., dendritic cells and macrophages) involved in virus transmission to helper T lymphocytes (CD4+ T cells), resulting in post-treatment infection relapse [49]. 

To address the limitations of cART and single ARV therapies, there is an urge to develop innovative strategies, such as nanocarriers for ARV delivery. Among the vast types of nanocarriers available, lipid-based nanocarriers can be one of the most attractive drug carrier classes for ARVs. The ability of these systems to transport drugs of varying lipophilicity, as well as their widely accepted biocompatibility and biodegradability, make them appealing for translation into clinical settings [50]. Additionally, as the oral route is the preferred method of administration, lipid nanocarriers stimulate the secretion of endogenous biliary lipids enhancing the GI absorptive capacity of the carried ARV agents [51]. Consequently, bioavailability enhancement and better distribution over the cellular and organ target viral sites are expected. Indeed, lipid nanocarriers can protect ARV agents (single or on association) through their body path, reducing accumulation in non-target tissues (reducing toxic side effects) and improving doses at viral reservoirs, as well as, avoiding unwanted drug interactions between the multiple carried drugs. The more specific and controlled delivery of ARV agents provided by lipid nanocarriers may also enhance therapeutic efficiency by decreasing the need for frequent administration regimens, which ultimately increases patient adherence. There is also the need to seek out novel alternatives capable of overcoming the physiological barriers inherent to oral ARV drug administration. Therefore, lipid nanocarriers are likewise advantageous to explore different non-invasive routes like skin transdermal, intranasal, and topical vaginal administration (for pre-exposure prophylaxis purposes) [19,52,53,54,55]. 

Nanocarriers for ARV drugs delivery in the CNS have also proved useful in circumventing the BBB because of their potential to enhance drug permeability. Nanocarriers have a variety of properties that help them penetrate the BBB and deliver drugs to the CNS, such as a high surface-to-volume ratio, a positive surface charge (to take advantage of adsorptive mediated transport through the BBB), and a small and regulated size (less than 200 nm) [47,56]. The charge and hydrophobicity of the nanocarriers’ surface impact plasmatic protein adsorption, and therefore their absorption and/or rate of transcytosis. Nanocarriers coated with specific surface stabilizers may be useful in achieving greater drug levels in the brain when it comes to CNS administration. Polysorbate 80 is a nonionic surfactant that has been shown to improve brain delivery in a variety of nanocarriers by adsorbing different apolipoproteins once in the circulation, imitating lipoproteins in their receptor-mediated transcytosis pathway into the CNS [47].

Given all of the mentioned advantages of nanocarriers for ARV delivery, the following section will provide a more detailed view of the types of lipid nanocarriers and their engineering properties to improve ARV therapy.

## 3. Lipid-Based Nanocarriers for Delivery of ARV Agents

Lipid-based nanocarriers are organic nanosystems that self-organize upon input of energy into a supramolecular structure with the hydrophilic portions (anionic, cationic, or zwitterionic) exposed to the surrounding aqueous solvent and the hydrophobic portions (usually hydrocarbon chains) facing each other to reduce contact with aqueous solvent [50]. Self-assembly is a common manufacturing method of lipid-based nanocarriers that is spontaneous but driven by an input of energy and the hydrophobic effect [50]. Lipid-based nanocarriers’ definitions and main characteristics are presented in Table 2.

Because of their nanoscale dimensions and adjustable surface properties, lipid-based nanocarriers are frequently advantageous for delivering antivirals to affected areas [67]. Several earlier studies have proven the potential of lipid-based nanocarriers to encapsulate and transport ARV. Representative examples are presented in Table 3, Table 4 and Table 5, highlighting the structure, composition, and physicochemical properties of nanocarriers, as well as the main challenges that were overcome by their use. Generally speaking, we may say that appropriate nanocarriers for drug delivery may be able to compensate for ARD drugs’ limitations and increase their pharmacological efficacy. One goal shared by all studies is the development of nanocarriers to reduce the side effects of ARD drugs. The concentration of AZT inside red blood cells, for example, caused hematopoietic toxicity. Liposomes containing AZT decreased blood cell absorption, overcoming AZT’s negative hematopoietic effects [57,68] (Table 3). Liposomes have also been shown to reduce ddI systemic exposure [69] while providing the same therapeutic effect as free NVP at lower doses (and thus with less toxicity) (Table 3) [70]. Furthermore, NLC containing EFV had lower toxicity when compared to the free drug (Table 5) [54]. Lipid nanocarriers can also be designed to improve drug bioavailability and prolong release thereby extending the dose interval. Because of their solid lipid matrix, SLN was the most capable of providing sustained release of the ARV encapsulated, reducing the frequency of administration. For example, the encapsulation in SLN of LPV, d4T, SQV, EFV, DRV, and AZT [64,71,72,73,74,75,76] provided a sustained release of this ARV even in the case of more hydrophilic drugs such as AZT (logP = 0.5) and d4T (logP = 0.72). All PI ARV (Figure 6) are cytochrome P450 3A substrates, which explains why most of them have poor pharmacokinetic features, such as extensive pre-systemic first-pass metabolism and short elimination half-lives [77]. There is also evidence that PI intracellular concentrations are influenced by P-gp and/or the activity of other efflux transporters. Moreover, some other ARV drugs inhibitors of RT have also documented short plasma half-lives (e.g., d4T, AZT, ddI) [78] which reduce their target tissue distribution. The ability of lipid nanocarriers to mediate the ARV distribution and increase their half-lives is therefore an important advantage. For example, liposomes have been documented as capable to increase half-lives of d4T, AZT, ddI, and RTV [13,14,15,16,20,79,80] (Table 3). Finally, many ARTs have a limited bioavailability in the brain, but the ability of lipid nanocarriers to mediate the brain delivery of ARVs has been widely documented, either for liposomes that potentially improve brain accumulation of AZT [81], or for SLN used for improving brain bioavailability of ATV, SQV, EFV, NVP and DRV [64,65,82,83,84,85], or NLC used as carriers of LPV, ATV, ETR [83,86,87] and NE improving brain accumulation of SQV and IDV [88,89,90]. From these studies it is worthwhile highlighting the SLN developed for EFV delivery that attained 150 folds more brain targeting delivery than the free drug [84]; the NLC for ATV delivery that attained 2.75 folds higher C_max_ at the brain and 4 folds higher brain bioavailability [86] and NE as a carrier of IDV that assured specific brain accumulation of the drug [89] (Table 3, Table 4 and Table 5).

Subsequently, a detailed and critical analysis of studies selected from Table 3, Table 4 and Table 5 is presented in Section 3.1 with a focus on the route of administration and Section 3.2 that focused on targeting strategies.

### 3.1. Tuning the Physicochemical Properties of Lipid-Based Nanocarriers to Overcome Biological Barriers

According to the administration route (Figure 7) and to achieve particularly in vivo performance and clinical applications, specific aspects of nanocarriers such as composition, size, surface charge, and coating need to be tuned [132,133,134,135,136]. 

Several studies have been conducted for parenteral administration of ARV (e.g., subcutaneous, intravenous). In this case, it is critical to extend the circulatory residence of the nanocarriers to ensure adequate time for distribution to the target tissues. Avoiding opsonization of serum proteins (e.g., human serum albumin, HSA) by controlling the size (<250 nm), charge (avoid positively charged nanocarriers), and surface coating with hydrophilic polymers (e.g., polyethylene glycol, PEG) are some strategies for extending circulation time. These criteria were met by Gagné et al. and Sudhakar et al. (liposomes) [80,94], by Heiati et al. (SLN) [18] and Pokharkar et al. (NLC) [54]. Aside from extending the circulation time of nanocarriers, it is also critical to use targeting strategies that can deliver ARV drugs to sites of latent HIV reservoirs such as lymph nodes, the spleen, and the gut mucosa, where HIV-target cells such as memory CD4+ T cells, macrophages, microglia, and astrocytes in the CNS are prevalent [137]. Some of these targeting strategies include: (i) surface functionalization of nanocarriers with sugar molecules like mannose [13,99] or galactose [14,15,96] that are recognized by lectin receptors found on the surface of cells from the mononuclear phagocyte system (MPS); (ii) coating of nanocarriers with hydrophilic molecules (e.g., amino acids, glucose) to facilitate BBB permeation by carrier-mediated transcytosis [55]; (iii) engineering of the lipid matrix of the nanocarriers (SLN, NLC, nanoemulsions) in order to mimic low-density lipoproteins (LDL) that are recognized by LDL receptors, thus facilitating BBB permeation by receptor-mediated transcytosis [54,55,64,65,85,87,89,90,120]; (iv) functionalization with ligands (e.g., HSA and monoclonal antibody (mAb)) that enhance BBB permeation by receptor-mediated transcytosis [64,120]; (v) inhibition of P-gp, which increases brain-specific accumulation [89]; and (vi) magnetic aided transport across BBB [115] and to MPS cells [116].

Oral administration is one of the preferred routes of administration due to its convenience that assures better adherence to the therapeutic regimens. However, this route of administration presents several limitations such as the variable absorption of the drugs, drug degradation by enzymes and acidic pH in the stomach, and first-pass metabolism effect. The physicochemical properties of drugs determine their absorption through the GI tract, namely their lipophilicity, which can be assessed by the logP. Typically, only drugs with logP values between 1 and 3 have favorable oral absorption profiles [138]. Most ARV drugs are outside this range (Table 3, Table 4 and Table 5), being either extremely hydrophilic (e.g., ddI [97] and AZT [121]) or highly lipophilic (e.g., LPV [66,71,83,109], RTV [110], EFV [16,75,129], SQV [90,126], EFZ [131] and ATV [86]). Therefore, lipid-based nanocarriers may help ARV drugs achieving a balanced lipophilic/hydrophilic nature. Additionally, lipid-based nanocarriers can be site-specific delivery systems by modifying their surface with ligands that are recognized at target tissues. For example, following oral administration, biotinylated liposomes of insulin were observed to permeate the GI tract via a facilitated absorption mechanism [139]. Based on this study, liposomes were coated with biotin (biotinylated proliposomes) to improve uptake of RTV into the intestinal lymphatic tissues [110]. Another example is the SLN grafting with a peptide that is specific for CD4+ receptors present on T cells, which improved specific DRV uptake by HIV host cells [82].

The nature of the components of nanocarriers also influences their functional performance when administered via a specific route. In the case of the oral route, the components of lipid-based nanocarriers induce the production of endogenous biliary lipids, which form colloidal structures in the presence of bile salts and significantly improve the solubilization and absorption capacity of ARVs in the small intestine [51]. Furthermore, the inclusion of penetration enhancers (e.g., Transcutol^®^ [66,127,129,131] and biliary salts (deoxycholic acid, sodium cholate) [90,125]) in the lipid matrix composition also improves the oral delivery of ARVs agents.

Transdermal administration, as opposed to oral administration, avoids the first-pass metabolism effect of drugs. As a result, a lower quantity of drugs can be administered efficiently by the transdermal route with reduced toxicity to achieve the same bioavailability as the oral route [72,113]. The fact that not all drugs can be delivered transdermally is one of the major drawbacks of this method. Drugs with a high molecular weight (>500 Da) cannot penetrate the stratum corneum [11]. In the pharmaceutical field, lipid-based nanocarriers are the most used for dermal/transdermal drug delivery. To improve skin permeation and efficiency, the composition of liposomes is changed to create new classes of lipid vesicles known as transferosomes, niosomes, ethosomes, cubosomes, and tocosomes. Jain et al. developed ethosomes that, due to the high amounts of ethanol, aid in breaking the stratum corneum and have higher elasticity, which contributes to improved 3TC skin permeation [112]. Chettupalli et al. produced cubosomes that improved ATV transdermal permeation due to the bioadhesive and permeation enhancer effect of their components [113]. SLN and nanoemulsions have also proved effective for the transdermal delivery of LPV [72] and AZT [128] respectively.

The vaginal administration is a promising route that allows self-administration of ARV drugs and permits achieving both local and systemic effects. In the case of local administration, the vaginal route avoids systemic exposure reducing side effects. If systemic administration is intended then drugs should have hydrophobic properties and low molecular weight [140,141]. The vaginal route may also be advantageous for drugs that undergo extensive metabolism, as it avoids the hepatic first-pass effect and allows for a reduction in the doses of drugs administered [141]. However, the vaginal route has been exclusively considered for topical pre-exposure prophylaxis (PrEP), as a preventative approach. Due to the unique characteristics of this mucosal site, administering ARV drugs via the vaginal route is a huge challenge because a fine-tuning of mucoadhesiveness/muco-penetration is required to ensure good distribution along the cervicovaginal lumen. In this regard, lipid nanocarriers can be used to improve ARV permeation into the vaginal mucosa, but there are some requirements in terms of size (>100, preferentially 200–500 nm) and surface charge (positively charged nanocarriers are mucoadhesive and hinder diffusion, whereas PEGylation promotes mucosal permeation) [142]. These requirements were considered in an in vitro study in which liposomal hydrogels were developed for the delivery of two ARV drugs with different lipophilicities [52]. As such, the hydrogel (hydrophilic) was used as a carrier for the hydrophilic drug FTC, while the liposomes were used as carriers for the more lipophilic drug TDF. The size and zwitterionic charge of the liposomes, as well as the hydrophilic nature of the gel, imply that there are fewer interactions with mucin from the mucosa, which may translate to higher drug diffusion [52]. SLN was also strategically developed to improve TFV uptake by virus-infected cells via vaginal administration [118]. TFV-loaded SLN were functionalized with a combination of peptide (PLL), to enhance intracellular uptake of the drug, and heparin, which can direct nanocarriers to killer lectin-like receptors of natural killer (NK) cells, resulting in direct killing of virus-infected cells [118]. Moreover, SLN possessed an adequate size and high density of negative surface charge that creates a hydrophilic surface that facilitates diffusion and minimizes entrapment into mucus [118]. In another study, a hybrid system composed of polymeric nanofibers containing liposomes loaded with FTC and TDF provided rapid onset of local ARV levels in mice after a single vaginal administration compared to five days of continuous daily use of oral TDF/FTC [117]. These results may be also translatable into a fairly wide protection time window in humans [117]. 

Intranasal administration has recently been investigated as a potential alternative to intravenous and other systemic administration routes for providing direct access to the brain via axonal transport along the olfactory nerve [50]. This administration route has the advantage of increased bioavailability due to the absence of first-pass liver metabolism and subsequent rapid absorption, resulting in a rapid therapeutic effect [50]. The disadvantages of this route are related to the limited amounts of drugs that can be delivered into the brain and to the mucociliary clearance mechanism that can remove toxic substances, drugs, nanocarriers, and microorganisms caught in the mucus layer [50]. To overcome the mucociliary clearance mechanism, the lipid matrix composition, and the surface chemistry of the nanocarriers have been explored for ARV delivery. Tuning the surface coating is important to guarantee enough mucoadhesion to avoid the rapid removal of lipid nanocarriers from the nasal mucosa [50]. On the other hand, it is also necessary to impart the nanocarrier surface with mucopenetrating properties to improve diffusion from the nose to the brain [50]. For example, Pokharkar et al. and Mahajan et al. used PEG coatings as amucopenetrating strategy for intranasal brain delivery of EFV [54] and SQVM [88], respectively. Other ARV drugs (SQV and EFV) benefited from nanocarriers composed of lipids with mucoadhesive properties (e.g., monoolein) [114] or fatty acids with mucopenetration properties [54,84]. The ability of nanocarriers composed of fatty acids to be flexible and pass through the opening of the olfactory epithelium has been attributed to the surfactant nature of fatty acids, which may disrupt the nasal membrane [50].

### 3.2. Targeting Anatomical and Cellular Reservoirs

As previously mentioned, lipid nanocarriers’ surfaces can be functionalized to improve their targeting selectivity [143]. The reticuloendothelial system contains galactose and lectin receptors and thus galactosylated [14,15,96] and mannosylated [13,99] liposomes target these receptors and have been utilized to deliver AZT, ddI, and d4T to the reticuloendothelial system. Functionalization of lipid nanocarriers with mAb, such as anti-HLA-DR that target follicular dendritic cells, B cells, and macrophages that express the HLA-DR is another strategy to achieve targeting specificity. For example, immunoliposomes functionalized with mAb resulted in increased IDV accumulation in mouse lymph nodes, with an area-under-the-curve that was 126-fold more than that of the free drug [94]. Liposomes can also be coated with recombinant soluble CD4 molecules [144,145], the Fab’ fragment of monoclonal antibody F105 [100], or the Fab’ fragment of anti-HLA-DR antibody [146], which all target gp120 on HIV-infected cells [143,145]. Besides sugars and mAb, PEG is also a surface targeting moiety that increases lipid nanocarriers at the lymph nodes [143,147].

In the case of BBB targeting the lipid-based nanocarriers have the potential to reduce efflux transporter binding by increasing brain accumulation. When ATV was encapsulated in solid lipid nanoparticles, its accumulation in a human brain microvessel endothelial cell line (hCMEC/D3) was greatly increased, indicating that this is a promising strategy for delivering ATV across the BBB [65]. When SQV was given by oil-in-water nanoemulsions synthesized with essential polyunsaturated fatty acid-rich oils, the maximal concentration and area-under-the-curve values in the brain were five- and threefold higher than the aqueous suspension [90]. Besides the lipid nanocarrier composition, it is advisable their surface functionalization with targeting ligands (e.g., Transferrin, and apolipoproteins) [47] that are recognized by BBB receptors and favor BBB transcytosis (examples of additional targeting ligands for BBB crossing can be consulted in [56]). Finally, one of the most tested cell-penetrating peptides is the HIV-1 Tat peptide. Certain sections of this peptide, known as protein-transduction domains, can help it migrate through biological membranes. The fusing of -galactosidase to the Tat peptide is required for BBB permeability, which is independent of transporters and receptor-mediated endocytosis [47]. Hence surface functionalization of lipid nanocarriers with Tat peptide can be an effective strategy for BBB crossing [47]. Glutathione is another peptide that is frequently utilized to achieve brain targeting. This endogenous tripeptide has antioxidant properties and plays a key function in intracellular metabolite detoxification. The ability of glutathione to increase ARV drugs delivery to the brain via liposomes has been demonstrated [47]. 

## 4. Biotechnological Advances in ARV Delivery

In the previous sections we have presented the classical therapy approach of HIV infections/AIDS based on the use of ARV drugs. Although cART can reduce HIV replication and postpone the onset of HIV infection/AIDS, viral mutagenesis is common and can lead to ineffective ARV therapy. New prospects of HIV treatment include biotechnological approaches combining pharmacological compounds and, in particular, genetic therapy, which uses RNA interference (RNAi). RNAi mechanisms are used in the context of gene therapy to modulate/silence the expression of genes involved in disease. Small interfering RNA (siRNA) operate within the RNAi pathway and have become the focus of recent therapeutic applications. Double- and long-stranded RNAs interact with a complex of proteins in the cytoplasm of cells, which is then cut into small double-stranded RNAs (19–21 nucleotides), known as siRNA, via Dicer enzymes. When siRNA enters the RNA-induced silencing complex (RISC complex), its double strands are separated into two single strands (antisense and sense), the antisense strand (guide) binds to a messenger RNA (mRNA) with a complementary sequence, and the mRNA target is degraded by non-RISC-complex endonucleases, halting the production of the abnormally encoded protein or enzyme [148] (Figure 8). 

The high potential of this strategy in comparison to others stems from the fact that when an appropriate siRNA is used, regular expression of any other gene implicated in other diseases is possible. The significance of studies developed by Fire and Mello [149] to discover RNAi cellular mechanisms was recognized with the Nobel Prize in Physiology or Medicine in 2006 [150].

Hence, siRNA-based therapeutics may offer a safer, effective, and longer-lasting approach that has demonstrated potential as a more personalized approach in the treatment of many diseases where enzyme activity is implicated, in which we may include HIV infections [151,152]. However, some obstacles must be solved before this therapeutic strategy can be used in clinical settings. These include improving delivery tactics and lowering costs. During the last decade, several research groups have worked on the topic of drug/nucleic acid co-delivery, mostly focusing on lipid-based nanocarriers. Indeed, the use of lipid-based nanocarriers, most commonly cationic charged liposomes or SLN, have several advantages, such as their ability to complex anionic nucleic acids and to protect RNAi from serum nucleases degradation and prolong blood circulation, which allows better distribution into the target tissues [148]. These systems are also essential for intracellular delivery, working as effective carriers for traversing the cytoplasmic membrane. For example, Kim et al. formulated a stabilized liposome for systemic administration of siRNA using a humanized mouse model to target lymphocyte function-associated antigen 1 (LFA-1), i.e., the predominant integrin found on all leukocytes. In vivo studies demonstrated a selective siRNA absorption by T cells and macrophages [153]. The opportunities for such continued innovation in formulating ARV drugs/nucleic acid co-delivery systems [148,153,154,155] will ensure continued research in this field, which should eventually lead to their clinical use.

ARV and siRNA co-delivery has been proposed as a promising biotechnological strategy for the treatment of Alzheimer’s disease (AD) in a context other than HIV therapy. The amyloid hypothesis has emerged as the dominant theory to explain the molecular pathogenicity of AD, following the identification of AβP as the plaque-forming peptide aggregated and accumulated in the brain, and amyloid-β precursor protein (APP) as the gene locus responsible for amyloid β-peptide (AβP) production [156]. Accumulated plaques cause hyperphosphorylation of the microtubule-associated protein tau, which aggregates to form neurofibrillary tangles (NFT), synaptic dysfunction, cell death, and, eventually, AD [156]. Despite evidence supporting the amyloid hypothesis, many clinical trials focusing on Aβ components have failed to produce any AD-modifying therapies [156]. It has recently been discovered that SGR retro-inserts novel genomic complementary DNA into neuronal genomes and becomes dysregulated in AD, producing numerous APP variant genes, transcripts, and AβP that would remain in the brain in various potential forms (e.g., plaques, fibrils, prions, and soluble products) and may not be recognized by specific Aβ-antibodies used in the therapeutic attempts to target AβP [156,157]. As a result, SGR provides a novel mechanism for explaining AD pathogenesis and the failures of Aβ-related clinical trials [156,157].

Human epidemiological data on 100,000 older HIV-infected patients (>=65 years old) revealed that the world population’s 10 percent prevalence of AD was not confirmed in these patients [156]. In fact, 1000 HIV-infected patients with AD were expected, but only one documented AD/HIV-infected case occurred [156]. This finding supports the notion that brain RT is involved in SGR, and its inhibition by ARV drugs or the silencing of its expression by siRNA is recently seen as a possible AD preventive and/or therapeutic intervention.

## 5. Conclusions

Notwithstanding the cART overall success, it continues to raise some serious concerns, and its effectiveness is hampered by some limitations such as ARV resistance mechanisms, prolonged treatment regimens, drug-drug interactions, toxicity effects, and pharmacokinetics issues. Therefore, innovative strategies such as lipid-based nanocarriers for ARV delivery appear to overcome physiological barriers. However, many issues must be addressed before we can reap the benefits of appropriate nanotechnology-based delivery systems that could improve ARV therapeutic outcomes. To begin with, most studies do not provide a thorough characterization of the lipid-based nanosystems developed or provide an incomplete or non-systematic formulation development methodology. Indeed, most studies do not consider the impact of nanosystems’ composition on: (i) ARV encapsulation and loading efficiency; (ii) nanosystems’ size and surface charge potential; (iii) nanosystems’ ability to completely release the entrapped bioactives; and, finally, (iv) nanosystems’ efficiency to deliver ARV to the virus reservoirs where their effects should be evaluated. Biotechnological applications of lipid nanocarriers loaded with anti-HIV therapeutics, such as the use of lipoplexes for siRNA delivery for AD, were also discussed. This promising bidirectional strategy helps ARV cross the BBB while halting SGR genome mutations that appear to be the cause of therapeutic AD failures.

## Figures and Tables

**Figure 1 pharmaceutics-13-01294-f001:**
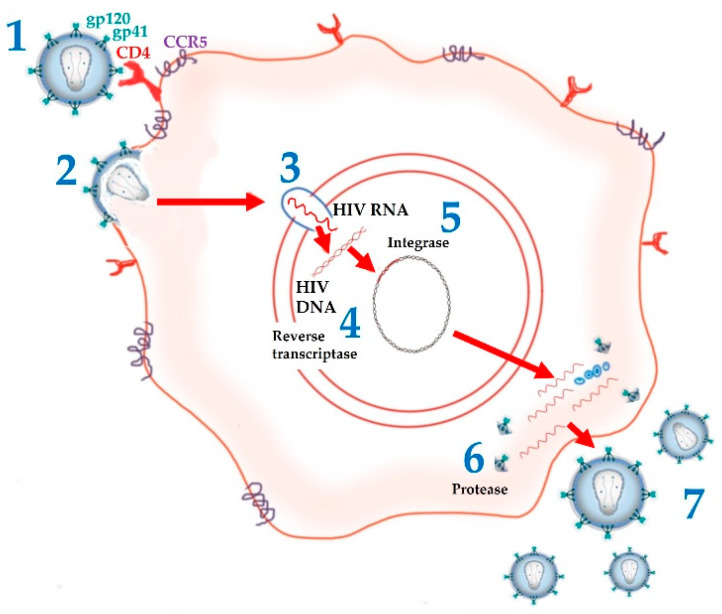
Stages in HIV lifecycle. (1) HIV attaches to CD4 receptor and CCR5 co-receptor. (2) HIV gp41 is exposed to the host cell and causes fusion. (3) HIV enters the nucleus and releases its enzymes and RNA. (4) Reverse transcriptase makes a double strand HIV DNA from HIV RNA. (5) Integrase includes HIV DNA in the DNA of the host cell. (6) New HIV viral components are produced, and Protease assembles new HIV virus. (7) Each host cell produces hundreds of new virions.

**Figure 2 pharmaceutics-13-01294-f002:**
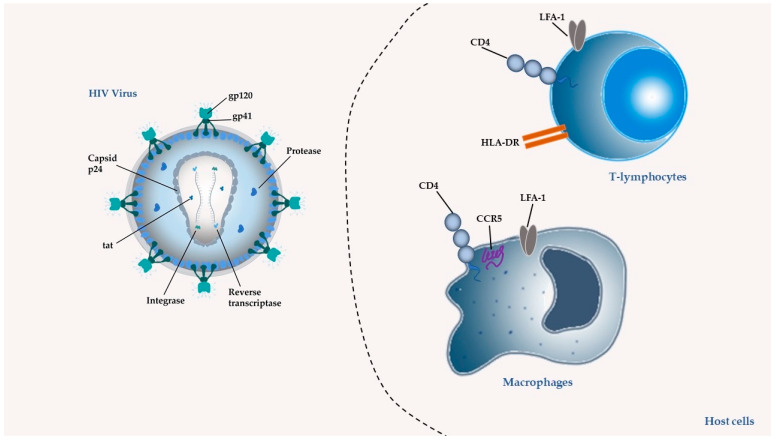
Molecular components of HIV virus and targets of ARV drugs. In the virus, the targets can be the glycoproteins responsible for adhesion gp120 and gp41; the enzymes integrase, reverse transcriptase, and protease; the protein from the capsid p24 and protein Tat that modulates transcription initiation and can reactivate a latently infected cell by penetrating. In the host cells the targets can be the lymphocyte function-associated antigen 1 (LFA-1); the CD4 receptor and its co-receptor C-C Motif Chemokine Receptor 5 (CCR5); and the human leucocyte antigen (HLA-DR).

**Figure 3 pharmaceutics-13-01294-f003:**
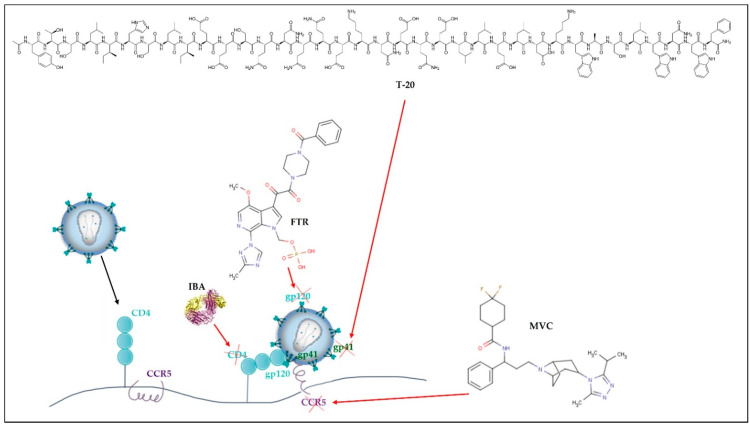
Cell entry inhibitors and fusion inhibitors. Ibalizumab-uiyk (IBA) blocks CD4 and maraviroc (MVC), blocks CCR5 receptors from host cells. Enfuvirtide (T-20) blocks gp41 and fostemsavir tromethamine (FTR) blocks gp120 from the virus.

**Figure 4 pharmaceutics-13-01294-f004:**
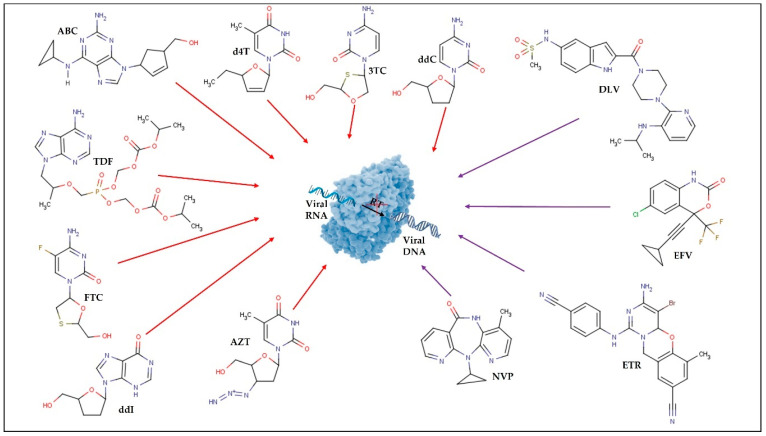
Reverse transcriptase (RT) inhibitors. Red arrows represent the nucleoside reverse transcriptase inhibitors (NRTI): lamivudine (3TC); abacavir (ABC); zidovudine (AZT); stavudine (d4T); didanosine (ddI); zalcitabine (ddC); emtricitabine (FTC); tenofovir disoproxil fumarate (TDF). Purple arrows represent the non-nucleoside reverse transcriptase inhibitors (NNRTI): efavirenz (EFV); etravirine (ETR); nevirapine (NVP); delavirdine (DLV).

**Figure 5 pharmaceutics-13-01294-f005:**
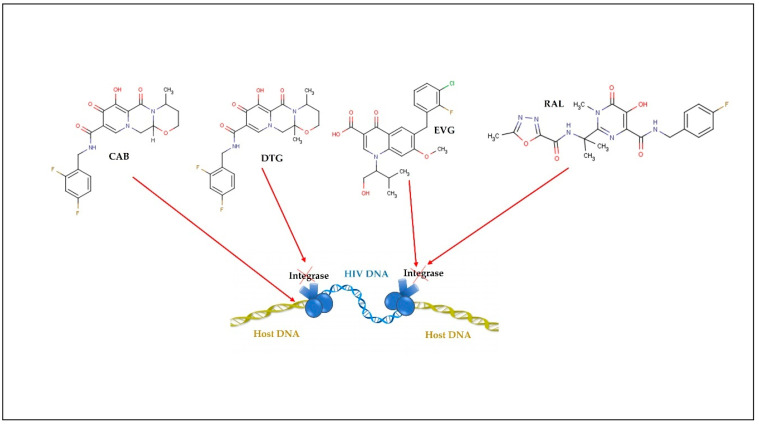
Integrase inhibitors: raltegravir (RAL); dolutegravir (DTG); elvitegravir (EVG); cabotegravir (CAB).

**Figure 6 pharmaceutics-13-01294-f006:**
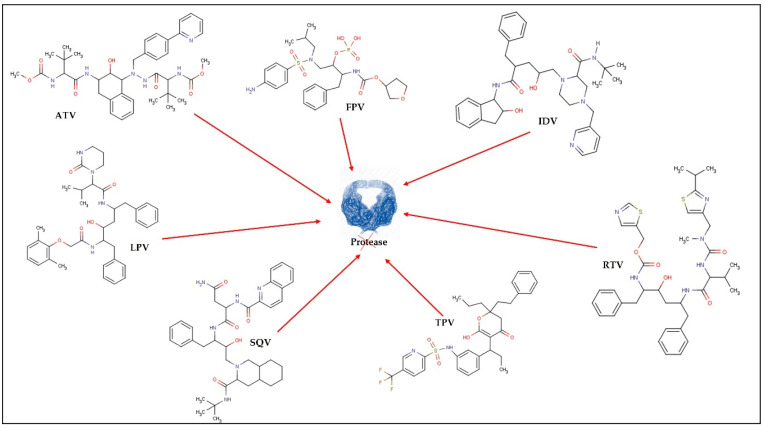
Protease inhibitors: tipranavir (TPV); indinavir (IDV); ritonavir (RTV); fosamprenavir (FPV); atazanavir (ATV); lopinavir (LPV); saquinavir (SQV).

**Figure 7 pharmaceutics-13-01294-f007:**
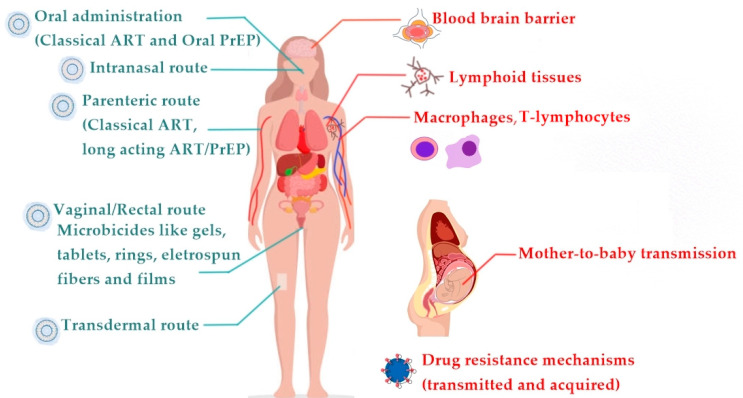
Potential ART (antiretroviral therapy) administration routes (left side in blue). The nanocarrier cartoon marks all possible routes of administration for lipid-based nanosystems. Classical ART formulations for therapy or pre-exposure prophylaxis (PrEP) are limited to oral and parenteral routes. Biological barriers to ART administration (right side in red).

**Figure 8 pharmaceutics-13-01294-f008:**
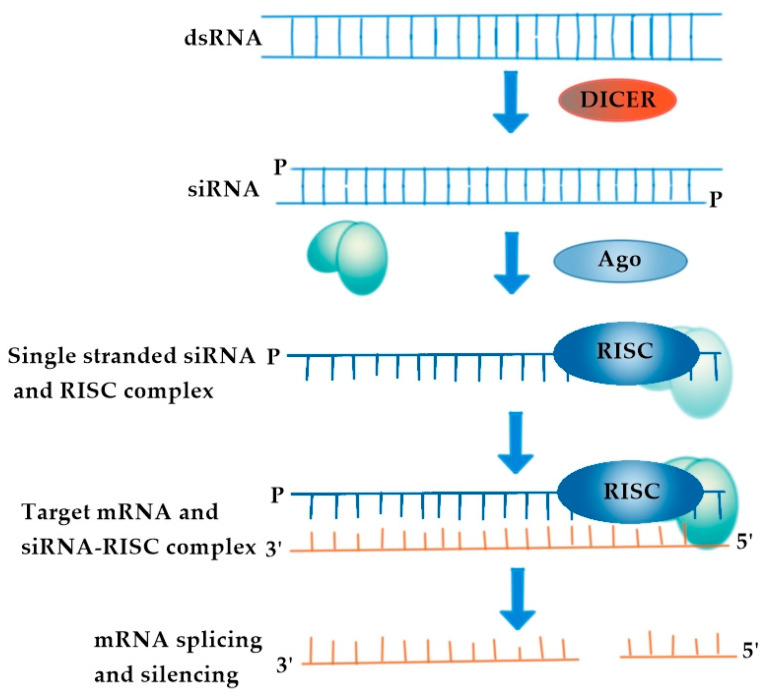
Mechanism of siRNA activity. RNA-induced silencing complex (RISC complex) is composed of Dicer and Argonaute protein (AGO).

**Table 1 pharmaceutics-13-01294-t001:** ARV agents are classified into therapeutic classes based on their mechanism of action and target site [31,33,34].

ARV Therapeutic Class		Mechanism of Action	ARV Single Agents and Some ARV Associations
Cell entry inhibitors	CCR5 antagonists	Block CCR5 coreceptors present on the surface of specific immune cells, preventing HIV from entering the cells.	MVC
	Attachment inhibitors	Bind to the gp120 protein on the viral outer surface, blocking HIV entry into CD4 cells.	FTR
	Post-attachment inhibitors	Block CD4 receptors present on the surface of specific immune cells, preventing HIV from entering the cells.	IBA
Fusion inhibitors (FI)		Interferes with HIV binding, fusion, and cell entrance by preventing the gp41 glycoprotein from being exposed to the virus-host cell membrane.	T-20
Nucleoside reverse transcriptase inhibitors (NRTI)		Block the viral RT, inhibiting HIV replication.	3TC; ABC; AZT; d4T; ddI; ddI EC; ddC (F.M.); FTC; TDF; 3TC+AZT; ABC+3TC; ABC+AZT+3TC; TDF+FTC
Non-nucleoside reverse transcriptase inhibitors (NNRTI)		Bind to viral RT and subsequently modify it, limiting HIV replication.	DOR; EFV; RPV; ETR; NVP; DLV
Integrase inhibitors (II)		Inhibition of viral integrase. Prevents the incorporation of HIV proviral DNA strands into the host cell genome.	RAL; DTG; EVG; CAB
Protease inhibitors (PI)		Inhibition of viral protease. Prevents the cleavage of some viral proteins and the maturation of virions, resulting in non-viral particles.	TPV; IDV; RTV; DRV; FPV; ATV; LPV+RTV; SQVM+RTV

Abbreviations: 3TC, lamivudine; ABC, abacavir; ATV, atazanavir; AZT, zidovudine; CAB, cabotegravir; CCR5, C-C chemokine receptor type 5; CD4, cluster of differentiation 4; d4T, stavudine; ddC, zalcitabine; ddI, didanosine; ddI EC, enteric coated didanosine; DLV, delavirdine; DNA, deoxyribonucleic acid; DOR, doravirine; DRV, darunavir; DTG, dolutegravir; EFV, efavirenz; ETR, etravirine; EVG, elvitegravir; FI, fusion inhibitors; FPV, fosamprenavir; FTC, emtricitabine; FTR, fostemsavir tromethamine; gp41, glycoprotein gp41; gp120, glycoprotein gp120; HIV, human immunodeficiency virus; IBA, ibalizumab-uiyk; IDV, indinavir; II, integrase inhibitors; LPV, lopinavir; MVC, maraviroc; NNRTI, non-nucleoside reverse transcriptase inhibitors (NRTI, nucleoside reverse transcriptase inhibitors; NVP, nevirapine; PI, protease inhibitors; RAL, raltegravir; RT, reverse transcriptase; RPV, rilpivirine; RTV, ritonavir; SQVM, saquinavir mesylate; T-20, enfuvirtide; TDF, tenofovir disoproxil fumarate; TPV, tipranavir.

**Table 2 pharmaceutics-13-01294-t002:** Lipid nanocarriers description, schematic representation, and main advantages and disadvantages for ARV delivery.

Type of Lipid Nanocarriers	Description and Main Characteristics	Advantages/Disadvantages for ARV Delivery	References
**LIPOSOMES** 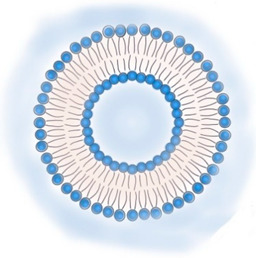 **ETHOSOMES** ** 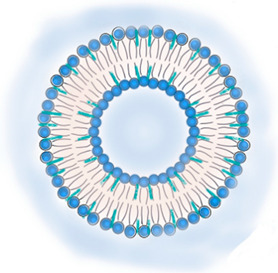 **	Aqueous phases in the core and surroundings of synthetic vesicles formed by self-assembly of lipid bilayers. Unilamellar (1 bilayer), oligolamellar (2–4 bilayers), and multilamellar (>4 bilayers) classifications are based on the number of lipid bilayers. Small (100 nm), large (100–500 nm), and giant (>500nm) are the classifications based on their size. Ethosomes are phospholipid-based vesicles with high ethanol content (20–45%).	Biocompatible and biodegradable. Administration routes are limited (mainly intranasal and intravenous). Production processes are difficult to scale. Liposomes in their natural state are quickly absorbed by the reticuloendothelial system and cleared from circulation. This property has been used to deliver ARVs to macrophages. The transdermal delivery of ARV is achieved by the incorporation of edge activators (e.g., surfactants, monoolein forming transferosomes) or ethanol (forming ethosomes) in the lipid bilayer. The protection of sensitive therapeutics can be achieved by using antioxidant agents in their composition (e.g., α-tocopherol, forming tocosome). Can encapsulate hydrophilic, hydrophobic, or amphiphilic drugs. Limited hydrophilic drug-loading capacity. Low long-term physical and biological stability, which hinders their use for long-term drug delivery.	[19,53,57,58]
**CUBOSOMES** ** 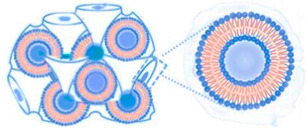 **	Highly stable structures organized in curved bicontinuous lipid bilayers forming soft 3D honeycomb-like structures. Composed by a continuous periodic bilayer and two non-connected water channels. Main components: glyceryl monooleate/monoolein (GMO) and phytantriol.	Incapability to modulate inner pore and channel sizes. Difficult loading of large molecules and difficult scale-up processes. Biocompatible and bioadhesive. Increase drug solubility and bioavailability through a variety of routes, including intranasal delivery to the brain and transdermal delivery. More stable than liposomes. High degree of encapsulation efficiency.	[59,60,61,62,63]
**LIPID NANOPARTICLES** 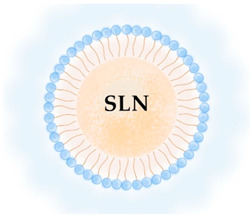 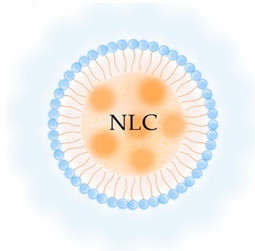	Colloidal self-assembled dispersions with a hydrophobic matrix and a surfactant layer that facilitates dispersion in water. At body and room temperatures, lipid nanoparticles are solid. Solid lipid nanoparticles (SLN) are lipid nanoparticles with hydrophobic matrices made up of solid lipids. Nanostructured lipid carriers (NLC) have lipid matrices with solid lipids and liquid lipids (oils).	Ease manufacturing and scale-up Low-cost and recognized as safe (GRAS) excipients, and biocompatibility. Greater drug stability and better control over drug-release kinetics than liposomes, cubosomes, and nanoemulsions. In comparison to other nanocarriers, they can entrap a greater amount of lipophilic drugs, but are inadequate for encapsulating hydrophilic and amphiphilic drugs. Good blood stability. Receptor-mediated transcytosis allows lipid nanoparticles to cross the BBB (targeting low-density lipoproteins receptors).	[7,64,65]
**LIPID EMULSIONS** 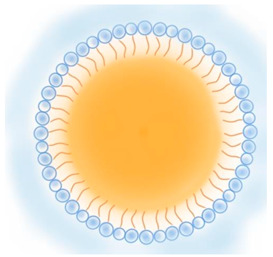 **Nanoemulsion O/W** 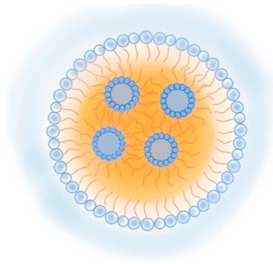 **SNEDDS W/O/W**	Colloidal systems made of immiscible liquid phases, categorized in water-in-oil (w/o) or oil-in-water (o/w), according to the phase dispersed in the other phase (continuous phase) and stabilized by surfactants. Microemulsions are thermo-dynamically stable dispersions that can be generated with low external energy. The droplet sizes of the dispersed phase are < 1000 nm, typically range between 10 and 200 nm, resulting in optically clear dispersion. Nanoemulsions are thermo-dynamically unstable and require high external energy to be produced. The dispersed phase droplets are < 500 nm typically 100 nm. Over time, nanoemulsions are more prone to instability. Self-emulsifying drug delivery systems (SEDDS) are emulsions that, when gently agitated, form fine oil-in-water droplets without the need for a dissolution process. These include self-micro emulsifying delivery systems (SMEDDS) with droplet sizes < 50 nm; self-nanoemulsifying drug delivery systems (SNEDDS) with droplet sizes of 20 to 200 nm; and solid self-nanoemulsifying oily formulations (S-SNEOF) where the drug is precipitated as a result of the evaporation of the co-solvent.	Increase drug oral bioavailability as their droplets preserve the drug from gastrointestinal degradation and can be dispersed quickly in blood and lymph (thereby avoiding the first-pass metabolism), but are also administrated by other routes: topical, and intravenous. Composed by GRAS lipids. However, to stabilize the droplets, high concentrations of surfactants are used, and thus their toxicity and biocompatibility may be compromised. Easy to manufacture and scale up, although the production methods can be expensive. In comparison to liposomes are more stable and provide higher encapsulation efficiency than lipophilic drugs. SNEDDS have higher physicochemical stability than classical nanoemulsions. SNEOFs promote lymphatic absorption by inhibiting first-pass metabolism and P-glycoprotein (P-gp) efflux, resulting in the complete eradication of HIV in lymphatic reservoirs.	[29,66]

Abbreviations: ARV, antiretroviral; BBB, blood-brain barrier; GI, gastrointestinal; GMO, glyceryl monooleate/monoolein; GRAS, generally recognized as safe; HIV, human immunodeficiency virus; NLC, nanostructured lipid carriers; o/w, oil-in-water; P-gp, P-glycoprotein; SEDDS, self-emulsifying drug delivery systems; SLN, solid lipid nanoparticles; SMEDDS, self-micro emulsifying delivery systems; SNEDDS, self-nanoemulsifying drug delivery systems; S-SNEOF, solid self-nanoemulsifying oily formulations; w/o, water-in-oil.

**Table 3 pharmaceutics-13-01294-t003:** Liposomes, ethosomes, cubosomes, and hybrid liposomal nanocarriers for ARV delivery.

Year	Composition	ARV	Physicochemical Characterization				Outcomes	Ref.
			E.E. (%)	D.L. (%)	ζ-Potential (mV)	Size (nm)		
**Liposomes**								
1990	DOPC:DPPG:Chol:Triolein (5:1:8:1)	ddC ^e^ logP = −1.35	48.3 ± 2.6	N.D.	N.D.	39,600 ± 11,900	↑ residence time in CNS ↑ controlled release	[91] *
1991/92	DPPC:DMPG (10:1) or DSPC:DMPG (10:1)	AZT ^a^ logP = 0.05	N.D.	LP w/DPPC: 9.35 ± 0.45 LP w/DSPC: 8.98 ± 0.56	N.D.	N.D.	↑ ARV activity ↓ hematopoietic toxicity	[57,68] *
1994	DSPC:DSPG (10:3)	ddI ^a^ logP = −1.24	85 ± 15	21 ± 1	N.D.	180 ± 20	↑ bioavailability ↓ systemic exposure ↓ effective fighting the virus compared to free ddI	[69] *^,^**
1995	DPPC:DCP:Chol (4:1:5)	ddC logP = −1.35	N.D.	35	N.D.	300	↑ ddC retention in macrophages	[92] **
1996	DSPC:DSPG (10:3) or DSPC:DSPG:DSPE-PEG (10:3:1.45)	ddI ^a^ logP = −1.24	N.D.	26 ± 4	N.D.	150 ± 10	Better pharmacokinetic profile ↑ viral reservoirs targeting ↑ bioavailability	[17]*^,^**
2000	EPC, DMPG, SM, DMPC, DPPC, DSPC, DMPC:Chol (4:1; 2:1 and 1:1), DSPC:Chol (4:1; 2:1 and 1:1), DPPC:Chol (2:1), DPPC:Chol:PS (6:3:1), DPPC:Chol: DCP (6:3:1)	d4T logP = −0.72	35 to 50	N.D.	+, − and neutral charges	600 to 1400	↑ maximal uptake by DPPC liposomes in macrophages and monocytes ↑ uptake with negative charged liposomes	[93] **
2002	DPPC:DPPG:DSPE-PEG-MAL (10:3:0.83)	IDV ^g^ logP = 2.9	11 ± 4	N.D.	N.D.	100 to 120	↑ [IDV] to lymphoid tissues	[94] *
2003	EPC:Chol (3:1)	IDV ^g^ logP = 2.9	97.5 ± 2.5 at pH 7.4 ≈ 20 at pH 5.5	19.5 ± 0.5 at pH 7.4 ≈4 at pH 5.5	N.D.	69 ± 7	↑ CD_4_^+^ T cells ↓ viral load in lymph nodes and plasma	[95] *
2005	SPC:AZT-M:V_E_ (6:2:0.1)	AZT-M ^a^	Liophilization Before: 99.4 ± 0.8 After: 98.3 ± 1.2	N.D.	- charges	Liophilization Before:88.5 ± 4.5 After:89.6 ± 6.3	↑ [ARV] in organs of RES and brain	[81] *
2006	EPC:Chol:SA (7:2:1) Uncoated LP or coated w/OPM: OPM-LP	d4T ^a^ logP = −0.72	Uncoated LP: 49.6 ± 1.2 OPM-LP: 47.2 ± 3.3	N.D.	Uncoated LP: + charge OPM-LP: charges ↓ close to neutrality	Uncoated LP: 120 ± 1.5 OPM-LP: 140 ± 2.3	↑ targeting ↑ residence time in HIV viral reservoirs ↑ d4T half-life ↑ pharmacological activity ↑ distribution ↓ elimination	[13] *^,^**
2007	EPC:Chol:DMPE (7:2:1) Uncoated LP or coated w/OPG: OPG-LP	d4T ^a^ logP = −0.72	Uncoated LP: 46.2 ± 0.69 OPM-LP: 47.1 ± 1.2	N.D.	Uncoated LP: +8.21 ± 0.15 OPG-LP: +3.2 ± 0.21	Uncoated LP: 122.3 ± 0.3 OPM-LP: 129.5 ± 0.3	↑ d4T half-life ↑ residence time ↑ hepatic cellular d4T uptake	[14] *^,^**^,^***
2008	EPC:Chol:DMPE (7:2:1) Uncoated LP or coated w/OPG: OPG-LP	d4T ^a^ logP = −0.72	Uncoated LP: 49.6 ± 1.2 OPG-LP: 48.7 ± 0.2	N.D.	N.D.	Uncoated LP: 120 ± 4 OPG-LP: 126 ± 4	Inhibition of HIV p24 protein with uncoated LP and OPG-LP ↑ accumulation of OPG-LP in the liver, spleen, and MPS ↓ uptake of OPG-LP in bone	[96] *^,^**
2006	EPC:Chol:PE (7:2:1) Uncoated LP or coated w/OPG: OPG-LP)	AZT ^a^ logP = 0.05	Uncoated LP: 54.3 ± 3.3 OPG-LP: 53.9 ± 2.1	N.D.	Uncoated LP: + charge OPM-LP: charges ↓ close to neutrality	Uncoated LP: 120.0 ± 2.1 OPG-LP: 136.9 ± 1.9	↑ AZT half-life ↑ residence time ↑ bioavailability	[15] *^,^**
2006	SPC:Span80^®^ (85:15) SPC:PEG-8-L (85:15)	AZT ^c^ logP = 0.05	LP w/Span80^®^: 63.5 ± 2.9 LP w/PEG-8-L: 57.1 ± 3.1	N.D.	LP w/Span80: −2.8 ± 0.4 LP w/PEG-8-L: −16.7 ± 0.7	LP w/Span80^®^: 132 ± 15 LP w/PEG-8-L: 116 ± 10	Better pharmacokinetic profile ↑ accumulation of AZT in target RES organs ↑ AZT half-life ↑ residence time, targeting, and controlled release	[19] *^,^**
2007	DPPC *Note: intended for oral administration*	ddI logP = −1.24	N.D.	N.D.	N.D.	1160 ± 129	↑ bioavailability	[97] **
2007	PC:POPG (3:1)	IDV logP = 2.9 SQV logP = 3.8	N.D.	N.D.	N.D.	130 to 150	↑ liposomal solubilization of both drugs ↑ drug concentration in the media (10- and 750-fold for IDV and SQV, respectively)	[98] **
2008	Plain-LP: SPC:PE:Span 80 (42.5:42.5:15) PEG-LP: SPC:PE:Span 80:MPEG 2000 (42.5:42.5:15:33.3)	AZT ^c^ logP = 0.05	Plain-LP: 63.5 ± 2.9 PEG-LP: 72.3 ± 4.5	N.D.	Plain-LP: −2.8 ± 0.4 PEG-LP: −18.2 ± 0.8	Plain-LP: 132 ± 14 PEG-LP: 158 ± 15	↑ cellular uptake in lymphoid cells ↑ biodistribution ↑ residence time and sustained drug release	[53] *^,^**
2008	LP: SPC:Chol (7:3) + charge LP: SPC:Chol:SA (7:3:1) - charge LP: SPC:Chol:DCP (7:3:1) w/Mannose: SPC:Chol:Man (7:3:2.5)	AZT ^g^ logP = 0.05	LP: 18.5 ± 1.2 + charge LP: 24.2 ± 0.9 − charge LP: 22.4 ± 1.4 Man-LP: 20.0 ± 2.5	N.D.	LP: +10.3 ± 1.8 +charge LP: +54.4 ± 2.3 −charge LP: −34.8 ± 4.45 Man-LP: +14.7 ± 3.9	LP: 122 ± 6 + charge LP: 126 ± 3 − charge LP: 128 ± 4 Man-LP: 127 ± 1.2	↓ release in Man-LP as compared to LP ↑ uptake ↑ localization of Man-LP in the lymph nodes and spleen	[99] *^,^**
2009	HSPC:Chol:mPEG–DSPE (55:40:5)	PI1	N.D.	N.D.	N.D.	N.D.	↑ and longer antiviral activity Facilitated specific uptake by non-phagocytic HIV-infected cells	[100] **
2010	EPC:Chol (9:1)	NVP logP = 2.5	78.1	7.81	N.D.	<200	↑ E.E. Quick *in vitro* release from liposomes	[101] **
2011	DPPC	ProddINP ^b^ logP = 0.05	99	8.83	−0.8 ± 0.5	187 to 208	↑ ddI blood half-life (3-fold) ↑ accumulation as prodrug at 24 h in various organs compared to plain drug	[79] *^,^**
2011	DPPC:EDPPC (1:1)	SFV logP = −19.5	N.D.	N.D.	N.D.	N.D.	Strong affinity of SFV for DPPC:EDPPC ↑ Affinity with ↑ cationic EDPPC Fusion w/viral/raft-mimicking vesicles	[102] **
2011	Chol:SA (194:1; 39:1; 22:1; 16:1;4:1) w/Span 20^®^/Span 40^®^/Span 60^®^	TFV logP = −1.6	3.46 to 65.26	N.D.	+4.79 to +17.13	36.13 to 114.9	The composition had a significant impact on TFV release Size and ζ were inversely proportional to the homogenization parameters, in contrast to the E.E. and conductivity TFV distributed within both the aqueous and lipid phases	[103] **
2012	EPC:DSPE-PEG	SQV logP = 3.8	32.2 ± 2.9	N.D.	−35.50 ± 1.66	176.6 ± 6.8	↓ cytotoxicity with PEGylated liposomes	[104] **
2012	DMPC:Chol:DPTAP (55:27:18) DPPC:Chol:DPTAP (55:27:18) DSPC:Chol:DPTAP (55:27:18) DSPC:Chol:SA (60:30:10)	TFV logP = −1.6	N.D.	N.D.	DMPC:Chol:DPTAP: +71.11 ± 5.72 DPPC:Chol:DPTAP: +62.50 ± 2.64 DSPC:Chol:DPTAP: +59.76 ± 2.49 DSPC:Chol:SA: +31.54 ± 1.90	DMPC:Chol:DPTAP: 166.8 ± 18.1 DPPC:Chol:DPTAP: 158.1 ± 32.0 DSPC:Chol:DPTAP: 159.0 ± 35.5 DSPC:Chol:SA: 158.5 ± 34.7	In the two-stage reverse dialysis method proposed, no drug leakage occurred during the 1st stage in LP containing high phase transition temperature lipids and high Chol content In the 2nd stage, significant differences in TFV release rate occurred in LP with different compositions	[105] **
2010/13	Chol:Phospholipon 100H:SA (1:1:0; 5:5:1; 3:3:1; 2.3:2.3:1; 2:2:1; 2:1:1)	TFV^.^ logP = −1.6	1.28 ± 0.24 (1:1:0) to 70.8 ± 2.55 (2:1:1)	0.39 ± 0.087 (1:1:0) to 17.71 ± 1.87 (2:1:1)	−3.43 (1:1:0) to +93.5 (5:5:1)	46.6 (1:1:0) to 2,200 (2:1:1)	↑ permeation of TFV (Caco-2 cell model)	[106,107] **
2016	LP DSPE:Stearic Acid:Chol (1:1:1) Stealth LP DSPE:Stearic Acid:Chol w/PEG 10000	RTV ^a^ logP = 3.9	LP: 98 ± 0.5 Stealth LP: 94.12 ± 0.29	LP: 11.92 ± 0.06 Stealth LP: 11.45 ± 0.03	LP: −33 ± 0.4 Stealth LP: −43.6 ± 1.8	LP: 49 ± 0.3 Stealth LP: 116.6 ± 0.1	Stealth LP prolongs RTV release to 34 h ↑ half-life of RTV for stealth LP LP and pure RTV showed dose dependent pharmacokinetics	[80] *,**
2017	Phospholipon 100H:Chol:SA (3:3:1 and 2:2:1) *Note: intended for oral administration*	TFV logP = −1.6	(3:3:1): 39.8 ± 8.1 (2:2:1): 68.1 ± 2.6	N.D.	+ charge	N.D.	↑ cellular permeability (10 times higher) ↑ E.E.	[108] **
2017	HSPC:Chol (7:3)	LPV ^b^ logP = 5.94	90.47 ± 0.32	N.D.	−24.8 ± 0.21	659.7 ± 23.1	↑ LPV release at 60 min (95% for LPV loaded proliposomes vs. 55% for free LPV) ↑ intestinal permeation (≈1.99 fold) compared to pure LPV) ↑ oral bioavailability (2.24- and 1.16-fold) than pure LPV and commercial LPV/RTV, respectively.	[109] *,**,***
2015	EPC:Chol:DSPE-PEG (9:1:1)	NVP logP = 2.5 and SQV logP = 3.8	NVP: 44 ± 2 SQV: 44 ± 1	N.D.	−29 ± 2	160 ± 2	↑ inhibition of viral proliferation at lower doses compared to free drugs NVP is mainly released in the early phases and SQV in the later phases of infection	[70] **
2017	SPC:Chol (2:1) Plain or coated w/biotin	RTV ^b^ logP = 3.9	Plain LP: 62.3 ± 1.7 Biotin-LP: 61.6 ± 1.8	N.D.	Plain LP: −18.9 ± 2.0 Biotin-LP: −26.1 ± 2.5	Plain LP: 126.6 ± 6.2 Biotin-LP: 149.8 ± 6.8	↑ release from biotin coated liposomes compared to conventional ones ↑ [RTV] in lymphatic tissues	[110] *^,^**
2018	DSPC:DSPE-mPEG2000 (9:1)	ATV ^g^ logP = 4.5 RTV logP = 3.9 TFV logP = −1.6	ATV: 99 ± 8.2 RTV: 92 ± 7.1 TFV: 10 ± 0.8	N.D.	N.D.	6 to 62	↑ residence time in plasma and peripheral blood mononuclear cells	[21] *
2019	DPPC *Note: intended for vaginal administration*	TDF logP = 2.65 FTC logP = −0.43	84	1	Zwitterio-nic	134 ± 13	↑ TDF permeation and ↑ sustained release Non-cytotoxic in CaSki (epidermoid cervical cancer cell line) and HEC-1-A (Human Endo-metrial Cancer-1)	[52] **
2020	POPC POPC:DPPE-PEG_2000_ (9:1)	T20 logP = −14.7 PPI xT20 + PPIX	N.D.	N.D.	Zwitterio- nic charge was predominantly affected by PPIX	Unloaded POPC: 110 nm Unloaded POPC:DPPE-PEG_2000_ (9:1): 120 nm Size was affected by PPIX	↑ entry inhibitors (T20 and PPIX) synergy compared to combination in free aqueous form	[111] **
**Ethosomes**								
2007	SPC w/ethanol	3TC ^c^ logP = −1.4	57.2 ± 4.1	N.D.	−8.2 ± 1.5	102 ± 13	↑ cellular uptake ↑ transdermal flux (25 times higher) ↑ elasticity contributes to enhanced skin permeation	[112] *^,^***
**Cubosomes**								
2021	GMO:CTAB:poloxamer 407(245:9:1, 219:9:1)	ATV ^g^ logP = 4.5	61 ± 4.6 (219:9:1) to 93 ± 1.2 (245:9:1)	N.D.	−29.41 (219:9:1) to −24.53 (245:9:1)	253 ± 5.6 (219:9:1) to 150 ± 8.7 (245:9:1)	↑ATV absorption and bioavailability (4.6 folds) compared to oral administration ↑ transdermal drug permeation due to bio-adhesive characteristic and permeation enhancement effect	[113] *^,^**^,^***
2020	GMO:CTAB: poloxamer 407 (18:15:1)	SQV ^b,d^ logP = 3.8	72 ± 2 higher concentrations of GMO favored drug entrapment	N.D.	N.D.	120 ± 2 ↑ particle size with ↑ GMO and ↓ Poloxamer 407	↑ SQV bioavailability (12-fold and 2.5-fold) when compared with oral and intranasal administration of free SQV	[114] *,***
**Hybrid liposomal nanocarriers**								
2017	SPC and gelatin nanoparticles (SG-LP)	d4T logP = −0.72	Gelatin NP (SG): 56.0 ± 1.7 SG-LP: 55.1 ± 2.1	N.D.	SG-LP: −44.6 ± 1.36	SG-LP: 232.9 ± 1.5	↑ controlled release ↑ uptake and hemocompatibility ↑ d4T half-life ↓ blood viremia	[20] **
2017	LP DPPC or DPPC:Chol (1:1, 4:1, 2:1) Magneto-plasmonic LP MNP@Au coated w/PEG	TDF logP = 2.65	↑ E.E. w/higher drug ratio (≈30% for LP:TDF (1:34)) ↑ E.E. w/smaller Chol content (≈60% for DPPC)	N.D.	N.D.	↓ with increasing Chol	↑ TDF release for LP without Chol ↑ transmigration across an in vitro BBB model by magnetic targeting ↓ viral replication of HIV infected microglial cells	[115] **
2010	LP SPC:Chol (1.2:1) Magnetic LP LP + magnetic AZTTP NP	AZTTP	54.5 ± 6	N.D.	N.D.	∼150 nm	↑ permeability (3-fold) for magnetic AZTTP LP than free AZTTP Efficient taken up by monocytes ↑ transendothelial migration in presence of an external magnetic field compared to normal/non-magnetic monocytes	[116] **
2021	LP DMPC:DOPE:Chol (7:2:1) inside PVA nanofibers	TDF ^h^ logP = 2.65 FTC logP = −0.43	100	4 (FTC) and 2.8 (TDF)	LP −0.67 ± 0.01	211 ± 24	Rapid onset of local drug levels upon single vaginal administration of fibers to mice comparing to the continuous daily use for 5 days of oral TDF/FTC Drug concentrations in vaginal fluids were fairly sustained up to 1–4 h, which could be translatable into a quite wide protection time window in humans	[117] *

Notes^: a^ intravenous injection; ^b^ oral administration; ^c^ transdermal administration; ^d^ intranasal administration; ^e^ intraventricular administration; ^g^ subcutaneous injection; ^h^ vaginal administration; ^N.D.^ no data * in vivo studies performed; ** in vitro studies performed; *** ex vivo studies performed. Abbreviations: 3TC, lamivudine; ARV, antiretroviral; ATV, atazanavir; Au, gold; AZT, zidovudine; AZT-M, zidovudine myristate; AZTTP, azidothymidine 5′-triphosphate; BBB, blood-brain barrier; CaSki, epidermoid cervical cancer cell line; Chol, cholesterol; CNS, central nervous system; CTAB, cetyltrimethylammonium bromide; d4T, stavudine; DCP, dicetyl phosphate; ddC, zalcitabine; ddI, didanosine; DMEM, Dulbecco’s Modified Eagle’s Medium; D.L., drug loading; DLMA, inner uncoated liposomes; DMPC, 1,2-dimyristoyl-sn-glycero-3-phosphocholine; DMPE, 1,2-dimyristoyl-sn-glycero-3-phosphoethanolamine; DMPG, 1,2-dimyristoyl-sn-glycero-3-phospho-(1′-rac-glycerol); DOPC, 1,2-dioleoyl-sn-glycero-3-phosphocholine; DOPE, 1,2-dioleoyl-sn-glycero-3-phosphoethanolamine; DPPC, 1,2-dipalmitoyl-sn-glycero-3-phosphocholine; DPPE-PEG_2000_, 1,2-dipalmitoyl-sn-glycero-3-phosphoethanolamine-N-[methoxy-(polyethylene glycol)-2000]; DPPG, 1,2-dipalmitoyl-sn-glycero-3-phospho-(1′-rac-glycerol); DPTAP, 1,2-dipalmitoyl-3-trimethylammonium-propane (chloride salt); DSPC, 1,2-distearoyl-sn-glycero-3-phosphocholine; DSPE, 1,2-distearoyl-sn-glycero-3-phosphorylethanolamine; DSPG, 1,2-distearoyl-sn-glycero-3-phospho-(1′-rac-glycerol); EDPPC, cationic 1,2-dipalmitoylethyl-phosphatidylcholine; E.E., entrapment efficiency; EPC, egg phosphatidylcholine; FTC, emtricitabine, Gal-DLMA, inner galactosylated liposomes; Gal-DMPE, galactosylated phosphatidylethanolamine; GMO, glyceryl monooleate; HEC-1-A, human endometrial cancer-1; HIV, human immunodeficiency virus; HSPC, hydrogenated soy phosphatidylcholine; IDV, indinavir; logP, partition coefficient; LP, liposome; LPV, lopinavir; MAL, maleimide; Man, mannose; MCZ, miconazole nitrate; MPEG 2000, mono methoxy PEG 2000; mPEG, methoxyl poly(ethylene glycol); MNP, magnetic nanoparticles; MPS, mononuclear phagocyte system; N.D., no data; NP, nanoparticles; NVP, nevirapine; OPG, O-palmitoylgalactose; OPM, O-palmitoylmannose; PBS, phosphate buffered saline; PC, phosphatidylcholine; PE, phosphatidylethanolamine; PEG, polyethylene glycol; PEG-8-L, octaoxyehtylene laurate ester; PLPC, 1-palmitoyl-2-lauroyl-sn-glycero-3-phosphocholine; POPC, 1-palmitoyl-2-oleoyl-sn-glycero-3-phosphocholine; POPE, 1-palmitoyl-2-oleoyl-sn-glycero-3-phosphoethanolamine; POPG, 1-palmitoyl-2-oleoyl-sn-glycero-3-phospho-(1′-rac-glycerol); PPIX, protoporphyrin IX; ProddINP, glycerolipidic prodrug of ddI; PS, phosphatidylserine; PVA, poly(vinyl alcohol); RES, reticuloendothelial system; RTV, ritonavir; SA, stearylamine; SFV, sifuvirtide; SM, sphingomyelin; SPC, soy phosphatidylcholine; SQV, saquinavir; T20, enfuvirtide; TDF, tenofovir disoproxil fumarate; TFV, tenofovir; V_E_, α-tocopherol.

**Table 4 pharmaceutics-13-01294-t004:** Lipid nanoparticles (SLN and NLC) for ARV delivery.

Year	Composition	ARV	Physicochemical Characterization				Outcomes	Ref.
			E.E. (%)	D.L. (%)	ζ-Potential (mV)	Size (nm)		
**SLN**								
1998	**Lipid phase:** Trilaurin:DPPC:DMPG (*0.69*:0.28: 0.03 wt:wt) **SLN coated w/PE-PEG_2000_** [10% (mol ratio)]	AZT ^a^ logP = 0.05	N.D.	N.D.	Plain SLN: −20 ± 5 PE-PEG SLN: −6 ± 4	Plain SLN: 183 ± 48 PE-PEG SLN: 182 ± 44	↓ release rate in SLN-PE-PEG ↑ bioavailability ↑ accumulation of SLN in the liver	[18] *^,^**
2006	**Lipid phase:** Trilaurin or Tristearin:Chol:PC:SA (1:0.5:1:0.1)	AZT ^a^ logP = 0.05	Trilaurin SLN: 57.8 ± 6.2 Tristearin SLN: 59.7 ± 6.1	N.D.	+charges	Trilaurin SLN: 130 ± 18 Tristearin SLN: 142 ± 22	↑ uptake in hepatocytes ↑ controlled release (12–15% in 24 h)	[76] *^,^**
2008	**Lipid phase:** stearic acid **Aqueous phase:** Pluronic^®^ F68 (3%) *Note: intended for enhanced brain delivery*	ATV logP = 4.5	98.9 ± 0.8 98.2 ± 1.3 89.3 ± 2.7	1 2 5	18.43 ± 0.70	167 ± 8.3	Burst ATV release of≈17% by 1 h and gradual release up to 40% by 24 h ↑ uptake and accumulation of ATV when delivered by SLN (human brain endothelial cell monolayer) compared to the free ATV	[65] **
2011	**Lipid phase:** Compritol^®^ 888 ATO/tripalmitin/cacao butter (wt of 8%) **Aqueous phase:** PC (7%), cholesteryl hemisuccinate (5%), taurocholate (2.5%) and 1-butanol (9.2%)	d4T logP = −0.72 DLV logP = 2.8 SQV logP = 3.8	SQV > DLV > d4T	N.D.	N.D.	142–308 ↑ % Compritol^®^ 888 ATO: ↑ d4T-SLN mean size and ↓ DLV-SLN and SQV-SLN mean size	↑ E.E. for d4T Sustained drug release: d4T > DLV > SQV	[74] **
2011	**Lipid phase:** Compritol^®^ 888 ATO **Aqueous phase:** Pluronic^®^ F127 (2.5%)	LPV ^b,f^ logP = 5.94	>99	N.D.	−26.5 ± 0.45	230.4 ± 5.6	Slow-release in both media pH 6.8 and pH 1.2 ↑ bioavailability and targeting ↑ % LPV secreted into the lymph	[71] *^,^**
2011	**Lipid phase:** Softisan^®^ 100 **Aqueous phase:** BSA and PAA (negative moiety) Additional layer of PLL (positively charged) and heparin (negatively charged) *Note: intended for topical/vaginal administration*	TFV logP = −1.6	8.3 ± 0.7	0.083	−51.07 ± 4.44	153.66 ± 11.33	Non-cytotoxic (human vaginal epithelial cell line) and easily functionalized ↑ solubility	[118] **
2011	**Lipid phase:** Dynasan^®^ 114 Solutol^®^ HS15 and Plurol^®^ Oleique CC 497 (hydrophobic surfactants 3%) **Aqueous phase:** Poloxamer 118 and Tween^®^ 80 (aqueous surfactant solution 0.5%)	d4T ^f^ logP = −0.72	96 ± 4.42	N.D.	−34.48	75 ± 1.22	↑ residence in splenic tissues ↑ uptake by macrophages compared to free drug	[119] *^,^***
2011	**Lipid phase:** Compritol^®^ 888 ATO/steric acid (4%) **Aqueous phase:** DODAB (1.8%), Tween^®^ 80 (1%), lecithin (0.2%) and 1-butanol (0.5%) **SLN coated with HSA**	NVP logP = 2.5	NVP-SLN: ≈77 (maximum achieved)	N.D.	NVP-SLN: + ↑ [HSA] ↓ ζ in SLN to values close to neutrality	NVP-SLN: 153.1 HSA/NVP-SLN: 189.2	↑ E.E. with SLN ↓ HBMECs viability with NVP-SLN ↑ HBMECs viability with HSA-NVP-SLN	[120] **
2013	**Lipid phase:** Dynasan^®^ 114/palmitic acid (4%) DSPE-PEG_2000_ (1.25%) *Note: wt fractions of palmitic acid in Dynasan-palmitic acid mixture were 0, 0.33, 0.67, 1* **Aqueous phase:** cholesteryl hemisuc-cinate (0.4%), poloxamer 407, Tween^®^ 80 and SDS (1%) *Note: wt fractions of poloxamer 407 in* *poloxamer 407-Tween*^®^ *80 were 0, 0.5, and 1, 0.1% (w/v)* *Note: intended for brain delivery* **mAb-grafted SQV-loaded SLN**	SQV logP = 3.8	≈55 to 80	N.D.	>−30	120 to 450	↑ BBB permeation ↑ bioavailability and ↑ solubility ↑ controlled release ↑ E.E. High biocompatibility of mAb-grafted SLN to HBMECs ↑ HBMECs uptake ↓ lymphatic uptake	[64] **
2014	**Lipid phase:** GMS **Aqueous phase:** Tween^®^ 80 (1.25%)	EFV ^b^ logP = 4.6	86	N.D.	−15.9	124.5±3.2	↑ 5.32-fold in C_max_ and ↑ 10.98-fold in AUC w/EFV-SLN compared to EFV suspension	[16] *
2015	**Lipid phase:** tristearin: HSPC:DSPE (11.2:12.6:0.3, molar ratio) **Aqueous phase:** Tween^®^ 80 (0.5%) **SLN conjugated with PA**	EFV ^a^ logP = 4.6	SLN: 72.1 ± 0.4 PA-SLN: 63.5 ± 0.6	N.D.	SLN: −28.8 ± 1.2 PA-SLN: −36.2 ± 1.0	SLN: 113 ± 0.2 PA-SLN: 163 ± 0.5	Able to permeate BBB ↑ bioavailability ↑ controlled release	[55] *^,^**
2015	**Lipid phase:** Gelucire^®^ 44/14 (30%) and Compritol^®^ 888 ATO (20%) **Aqueous phase:** Lipoid^®^ S 75 (25%) and poloxamer 188 (5%)	EFV ^b^ logP = 4.6	85.6	39.4	−35.55	168.92 ± 31.2	Prolonged and biphasic in PBS pH 6.8 ↑ bioavailability ↑ [EFV] in the spleen ↑ biodistribution in lymphatic organs	[75] *^,^**
2016	**Lipid phase:** stearic acid **Aqueous phase:** PVA **SLN modified with Aloe Vera**	AZT logP = 0.05	SLN: 66.5 SLN-AV: 84	SLN: 18.01 SLN-AV: 29.6	AZT-SLN: −12.18 to 13.1 AZT-SLN-AV: −14.2 to 15.41	SLN: 222 to 227 SLN-AV: 402 to 434	↑ solubility ↑ cellular uptake and no cytotoxicity (C6 glioma cells) ↑ E.E.	[121] **
2017	**Lipid phase:** GC and GMS (1.5 g) *Note: N.D. of GC:GMS* Span 80 (1%) **Aqueous phase:** Tween^®^ 80 (2%)	DRV logP = 1.89	SLN: 74.23 Freeze-dried SLN: 69.8	Post-freeze drying: 9.37	Freeze-dried SLN: −22 ± 2	SLN: 210 Freeze-dried SLN: 270	Sustained release of DRV until 12 h Apparent permeability across rat intestine: 24 × 10^−6^ (cm/s) at 37 °C and 5.6 × 10^−6^ (cm/s) at 4 °C Endocytic uptake	[73] **^,^***
2017	**Lipid phase:** tripalmitin **Aqueous phase:** poloxamer 188 (1%)	EFV ^d^ logP = 4.6	64.9	N.D.	−21.2	108.3	Burst release followed by a prolonged release ↑ [EFV] in brain ↑ brain targeting efficiency (more 150 times) and better absorption of the EFV (70 times more) with intranasal as compared to orally administered marketed formulation (capsule)	[84] *^,^**
2017	**Lipid phase:** GMS:SL (1:1) **Aqueous phase:** Poloxamer118 (0.5%) or Tween^®^ 80 (0.25%)	RTV logP = 3.9	21.4−53.3	N.D.	−39.35 ± 1.2 to −50.80 ± 4.8	178 to 254	↑ E.E. and mean size using poloxamer 118 as surfactant ↑ controlled release RTV-SLN can maintain inhibition of virus production	[122] **
2017	**Preconcentrate:** PGDS or stearic acid and poloxamer 118 (0.1–1.0%) Diluted with water	NVP ^a^ logP = 2.5	>90	N.D.	PGDS-SLN: −26.8 ± 2.1	Stearic acid-SLN: 940.2 ± 1.54 PGDS-SLN: 70 to 1100 (average particle size ~212 nm)	Biphasic release profile with an initial burst release ↑ [NVP] in the liver, kidneys, and brain Targeting for multiple viral reservoirs	[85] *^,^**
2018	**Lipid phase:** Compritol^®^ ATO 888 (0.5%) **Aqueous phase:** poloxamer 407 (0.25%), Labrasol^®^ (0.25%) **Topical formulation for transdermal delivery** SLN based HG	LPV ^c^ logP = 5.94	69.78	N.D.	−17.7 ± 0.54	48.86 ± 4.6	↑ sustained LPV release from SLN based HG (71.197 ± 0.006% after 12 h) compared to plain HG of the drug released (98.406 ± 0.007% after 4 h) SLN based HG resulted in the highest C_max_ (20.3127 ± 6056 μg/mL) compared to plain HG (8.0655 ± 1.6369 μg/mL) and oral LPV suspension (4.2550 ± 1 6.380 μg/mL)	[72] *^,^**^,^***
2018	**Lipid phase:** hydrogenated castor oil (castor wax) **Aqueous phase:** sodium oleate (3.5%) **Grafted SLN:** peptide (100 μg):SLN (1 μM) (Pept-DRV-SLN)	DRV ^b^ logP = 1.89	DRV-SLN: 90.10 ± 1.15 Pept-DRV-SLN 90.16 ± 1.25	DRV-SLN: 13.06 ± 1.18 Pept-DRV-SLN 13.18 ± 1.23	DRV-SLN: −50.1 ± 1.17 Pept-DRV-SLN −35.45 ± 1.10	D-SLN: 189.45 ± 2.10 Pept-D-SLN 195.11 ± 1.53	↑ DRV release in SLN compared to a plain drug suspension ↑ permeability in Caco-2 cells (4-fold) than free drug ↑ uptake in HIV host cells (molt-4 cells were taken as a model containing CD4 receptors) as compared to non-CD4 receptor-bearing Caco-2 cells ↑ bioavailability than free DRV: ↑ uptake in various organs (also in HIV reservoirs like spleen and brain) with Pept-DRV-SLN ↑ binding with the HIV host cells	[82] *^,^**
**NLC**								
2011	**Lipid phase:** OA (1%) and Compritol^®^ 888 ATO/steric acid (4%) **Aqueous phase:** DODAB (1.8%), Tween^®^ 80 (1%), lecithin (0.2%) and 1-butanol (0.5%) **NLC coated with HSA**	NVP logP = 2.5	NVP-NLC: ≈68 (maximum achieved)	N.D.	NVP-NLC: + ≈+17.5 ↑ [HSA] ↓ ζ of NLC to values close to neutrality	NVP-NLC: 159.6 HSA/NVP-NLC: N.D.	NLC promote a fast release	[120] **
2017	**NLC** **Lipid phase:** Compritol^®^ ATO 888:OA (1:3) Tween^®^ 80 (44 mg) **MLN** **Lipid phase:** Compritol^®^ ATO 888:OA:Span^®^ 80 (1:3:1.8) Tween^®^ 80	3TC^.^ logP = −1.4	3TC-NLC: 34 ± 1 3TC-MLN: 20 ± 2	3TC-NLC: 0.3 ± 0.01 3TC-MLN: 1.08 ± 0.06	Unloaded-NLC: −42.9 ± 0.7 3TC-NLC: −45 ± 2 Unloaded MLN: −24.5 ± 0.4 3TC-MLN: −21 ± 2	Unloaded-NLC: 229 ± 2 3TC-NLC: 218 ± 4 Unloaded MLN: 426 ± 9 3TC-MLN: 450 ± 10	Sustained and controlled 3TC release under gastric and plasma-simulated conditions (≈45 h in MLN) Low cytotoxicity (T lymphocytes) for both formulations ↑ loading capacity and storage stability with MLN	[123] **
2018	**Lipid phase:** Precirol^®^ATO 5:Miglyol^®^ 812 (3:1) Tween^®^ 80 (158 mg) AZT-NLC prepared by hot ultrasonication method and M-AZT-NLC prepared by the one-step microwave-assisted method	AZT logP = 0.05	AZT-NLC: 44 ± 3 M-AZT-NLC: 22 ± 2	AZT-NLC: 0.31± 0.04 M-AZT-NLC: 1.41 ± 0.02	- AZT-NLC: −29 ± 2 M-AZT-NLC: −20 ± 1	AZT-NLC: 266 ± 4 M-AZT-NLC: 113 ± 3	Controlled release of AZT Suitable for oral administration	[12] **
2019	**Lipid phase:** Compritol^®^ 888 ATO:OA (3.7:1) **Aqueous phase:** Tween^®^ 80 (0.04)	LPV ^b^ logP = 5.94	83.6	N.D.	+ 21.2	196.6	↑ bioavailability ↑ [LPV] in the brain ↑ uptake and ↓ cytotoxicity (Caco-2 cells and macrophages)	[83] *^,^**
2020	**Lipid phase:** Precirol^®^ ATO 5:Lauroglycol^TM^ 90 (70:30) Cremophor^®^ RH 40 (3%)	ATV ^b^ logP = 4.5	71.09 ± 5.84	8.12 ± 2.7	−11.7 ± 0.47	227.6 ± 5.4	Fast release (60%) in the initial 2 h, followed by sustained release ↑ permeation of ATV (2.36-fold) across the rat intestine as compared to the free drug 2.75-fold greater C_max_ in the brain and a 4-fold improvement in brain bioavailability as compared to the free drug	[86] *^,^**^,^***
2020	**Lipid phase:** Monosteol^TM^ (71.5%): Capmul^®^ PG 8 (28.5%) **Aqueous phase:** Tween^®^ 80 (0.43%) and poloxamer 188 (1.3%) **Cationic NLC** **Lipid phase:** same composition **Aqueous phase:** same composition + CTAB (1% w/w of lipid phase)	EFV logP = 4.6	EFV-NLC: 91.18 ± 2.9 Cationic EFV-NLC: 90.21 ± 2.3	EFV-NLC: 10.94 ± 0.35 Cationic EFV-NLC: 11.04 ± 0.17	Plain NLC: −15.16 ± 0.69 EFV-NLC: −15.8 ± 1.21 Cationic EFV-NLC: +23.86 ± 0.49	Plain NLC: 114.53 ± 5.63 EFV-NLC: 116.5 ± 9.59 Cationic EFV-NLC: 105.6 ± 4.93	↑ solubility Excellent cytocompatibility (CC_50_ 13.23±0.54 µg/mL) Uptake of cationic NLC by THP-1 macrophages ↑ retention/sustained release and ↑ inhibition of HIV-1 (2.32-fold) in infected macrophages with cationic NLC compared to the free drug ↑ anti-HIV-efficacy (2.29-fold) with cationic NLC	[124] **
2021	**Lipid phase:** Precirol^®^ ATO 5:Capmul^®^ MCM (40:60) Capryol^TM^ 90 (N.D.) **Aqueous phase:** Lutrol^®^ F 127 (1%)	ETR ^a^ logP = 4.5	>90	5 to 10	−20 ± 2.3	351.7 ± 3.36	↑ cellular uptake and ↑ anti-HIV efficacy Overall better pharmacokinetics as compared to the free drug ↑ [ETR] several-fold in the liver, ovary, lymph node, and brain as compared to the free drug	[87] *^,^**
2021	**Lipid phase:** GMS/Gelucire^®^ 50/13/Dynasan^®^ 118:Capmul^®^ MCM EP (80:20) (8%) Span^®^ 80 **Aqueous phase:** Tween^®^ 80, sodium cholate; PEG 6000 (1%), propylene glycol (1%), BHT (0.4%) *Note: Surfactant mixture [(Tween*^®^ *80: Span*^®^ *80 (70:30)]: 5%*	DTG logP = 2.2	88.09	N.D.	−16.6	123.1	Sustained release over 48 h ↑ DTG permeation through rat intestine (≈94.02%) as compared to plain drug suspension (only 55.62%) after 8 h	[125] **^,^***
2014	**Lipid phase:** Precirol^®^ ATO 15 (10%) and Miglyol^®^ 812 (1%) **Aqueous phase:** Tween^®^ 80 (1%) and poloxamer 188 (1 or 0.5%) **Coating on NLC:** Dex–Prot	SQV logP = 3.8	All formulations: 99	N.D.	Uncoated NLC: −36 ± 6 to −22 ± 4 Dex-Prot NLC: −0.5 ± 4 to +12 ± 4	Uncoated NLC: 152 ± 1 to 936 ± 1 Dex-Prot NLC: 244 ± 1 to 1326 ± 1	↑ permeability (up to 9-fold) with Dex–Prot NLC in comparison to uncoated NLC (Caco-2/HT29-MTX co-culture monolayer model)	[126] **
2017	**Lipid phase:** Precirol^®^ ATO 5:Captex^®^ P 500 (7:3) **Aqueous phase:** MYS-25 (2%)	EFV ^a,d^ logP = 4.6	95.78 ± 0.42	N.D.	−18.7 ± 1.0	161 ± 2.8	EFV release of 92.45% after 24 h The therapeutic concentration of EFV in the CNS following intranasal administration No toxicity of encapsulated EFV compared to free EFV	[54] *^,^**

Notes: ^a^ intravenous injection; ^b^ oral administration; ^c^ transdermal administration; ^d^ intranasal administration; ^f^ intraperitoneal administration; ^N.D.^ no data * in vivo studies performed; ** in vitro studies performed; *** ex vivo studies performed. Abbreviations: 3TC, lamivudine; ARV, antiretroviral; ATV, atazanavir; AUC, area under the curve; AV, aloe vera; AZT, zidovudine; BBB, blood-brain barrier; BHT, butylated hydroxy toluene; BSA, bovine serum albumin; Capmul^®^ MCM EP, glycerol monocaprylocaprate; Capmul^®^ PG 8, propylene glycol monocaprylate; Capryol^TM^ 90, propylene glycol monocaprylate; Captex^®^ P 500, triglycerides and esters prepared from fractionated vegetable oil sources and fatty acids from coconuts and palm kernel oils; CC_50_, concentration at which 50% cells are viable; Chol, cholesterol; C_max_, maximum concentration; CNS, central nervous system; Compritol^®^ 888 ATO, glycerol dibehenate; Cremophor^®^ RH 40, polyoxyl 40 hydrogenated castor oil; CTAB, cetyltrimethylammonium bromide; d4T, stavudine; Dex–Prot, dextran–protamine; D.L., drug loading; DLV, delavirdine; DMPG, 1,2-dimyristoyl-sn-glycero-3-phospho-(1′-rac-glycerol); DODAB, dioctadecyl dimethylammonium bromide; DPPC, 1,2-dipalmitoyl-sn-glycero-3-phosphocholine; DRV, darunavir; DSPE, 1,2-distearoyl-sn-glycero-3-phosphorylethanolamine; DTG, dolutegravir sodium; Dynasan^®^ 114, trimyristin; Dynasan^®^ 118, glyceryl tristearate; E.E., entrapment efficiency; EFV, efavirenz; ETR, etravirine; GC, glyceryl caprylate; Gelucire^®^ 44/14, lauroyl polyoxyl-32 glycerides; Gelucire^®^ 50/13, stearoyl polyoxyl-32 glycerides; GMS, glyceryl monostearate; HBMECs, human brain microvascular endothelial cells; HG, hydrogel; HSA, human serum albumin; HSPC, hydrogenated soy phosphatidylcholine; Labrasol^®^, caprylocaproyl polyoxyl-8 glycerides; Lauroglycol^TM^, 90 propylene glycol monolaurate; Lipoid^®^ S 75, fat free soybean phospholipids with 70% PC; LPV, lopinavir; mAb, 83-14 monoclonal antibody; Miglyol^®^ 812, medium-chain triglycerides; MLN, multiple lipid nanoparticles; Monosteol^TM^, palmitate/stearate of propylene glycol; MYS-25, polyethylene glycol 25 stearate; N.D., no data; NLC, nanostructured lipid carrier; NVP, nevirapine; PA, phenylalanine; PAA, poly(acrylic acid); PC, phosphatidylcholine; PE-PEG2000, dipalmitoylphosphatidylethanolamine-N-[poly(ethylene glycol)2000]; PEG, polyethylene glycol; Pept-DRV-SLN, peptide grafted-darunavir loaded SLN; PGDS, polyglyceryl-6-distearate; PLL, poly(L-lysine hydrochloride; Plurol^®^ Oleique CC 497, polyglyceryl-3 dioleate; Precirol^®^ ATO 15, glyceryl palmitostearate; PVA, poly vinyl alcohol; RTV, ritonavir; SA, stearylamine; SDS, sodium dodecyl sulfate; SL, soy lecithin; SLN, solid lipid nanoparticle; Softisan^®^ 100, hydrogenated coco-glycerides; Solutol^®^ HS15, polyoxyl 15 hydroxystearate; SQV, saquinavir; TFV, tenofovir; w/w- weight/weight; *wt*—weight.

**Table 5 pharmaceutics-13-01294-t005:** Lipid emulsions (microemulsions, nanoemulsions, and self-emulsifying drug delivery systems) for ARV delivery.

Year	Composition	ARV	Physicochemical Characterization				Outcomes	Ref.
			E.E. (%)	D.L. (%)	ζ-Potential (mV)	Size (nm)		
**Microemulsions (ME)**								
2015	**o/w ME** **Lipid phase:** Capmul^®^ MCM (75%), Cremophor^®^ RH 40 **Aqueous Phase** Transcutol^®^ P *Cremophor RH 40: Transcutol*^®^ *P (1:1)(40%)* **Solid ME** Absorbing agent (aerosil 200) ME:aerosil 200 (1:1)	DRV logP = 1.89	99.42	N.D.	N.D.	40.68	↑ solubilityGreater intestinal permeability than the free drug ↑ intestinal permeability with ↑ [oil phase]	[127] ***
2016	**Lipid phase:** isopropyl myristate (10%), Labrasol^®^ (30%); Oleic Plurol^®^ (10%)	AZT ^c^ logP = 0.05	N.D.	N.D.	N.D.	N.D.	↑ AZT permeated (≈2-fold) as compared to control—HGNo apparent skin irritation; little histological changes in mice skin	[128] *^,^**
**Nanoemulsions (NE)**								
2008	**o/w** **Lipid phase:** Flax-seed oil or safflower oil (1 mL) **Aqueous phase:** EPC (3%) and deoxycholic acid (1%)	SQV ^a,b^ logP = 3.8	N.D.	N.D.	- SQV-Flax-seed NE: −43.28 ± 3.79 SQV-Safflower NE: −49.55 ± 5.02	SQV-Flax-seed NE:218.0 ± 13.9SQV-Safflower NE:140.0 ± 12.6	↑ SQV (3-fold) in systemic circulation when loaded in Flax-seed NE than in the free form↑ bioavailability↑ brain distribution (C_max_ 5-fold and AUC 3-fold) higher in the brain with Flax-seed NE than the free drug	[90] *
2014	**NE** **Lipid phase:** Capryol^®^ 90, Geucire^®^ 44/14 (13.728%) **Aqueous phase:** Transcutol^®^ HP (3.432%) and water (79.98%) **Oil:Smix (1:6)**	EFV ^b^ logP = 4.6	N.D.	N.D.	N.D.	26.427 ± 1.960	>80% release within 6 h ↑ AUC_0__→__24h_ (43.53 μg h/mL) compared to EFV suspension (20.65 μg h/mL)EFV absorption resulted in 2.6-fold increase in bioavailability in comparison to the free EFV	[129] *^,^**
2014	**o/w NE** **Lipid phase:** Capmul MCM (6%) **Aqueous phase:** Tween^®^ 80 (6%), PEG 400 (2%) and water (86%)	SQVM ^d^ logP = 3.8	96.8 ± 1.2	N.D.	−10.3 ± 1.67	176.3 ± 4.21	↑ Diffusion of SQVM across nasal mucosa than the free drug No significant adverse effect in cilia toxicity study ↑ [SQVM] brain after intranasal administration of NE than intravenous delivery of free drugEffective CNS targeting	[88] *^,^**
2013	**o/w NE** **Lipid phase:** soya bean oil (10%), Chol (0 or 0.3%), EPC-80 (1.2%), α-tocopherol (0.25%) OA (0.3%) **Aqueous phase:** glycerol (2.25%), Tween^®^ 80 (0 to 1%), and double-distilled water (10%)	IDV ^a^	No Chol, no Tween^®^ 80: 99.1 ± 0.2 Chol and no Tween^®^ 80: 98.9 ± 0.03 Tween^®^80 (1%): 98.97 ± 0.2	N.D.	No Chol and no Tween 80 NE: −35.8 ± 6.04 Chol and no Tween 80 NE: −31.3 ± 1.80 Tween 80 (1%) NE: −40.1 ± 4.05	No Chol and no Tween 80 NE:329.5 ± 3.08 Chol and no Tween 80 NE:237.0 ± 5.08 Tween 80 (1%) NE:196.0 ± 3.54	NE containing Chol and higher [Tween 80] (1%) had lower globule size, relative better release, and higher ζ↑ brain uptake of IDV in Tween 80 (1%) NE compared to Chol and no Tween 80 NE↑ brain level of IDV administered by Tween 80 (1%) NE compared to the free drug (2.44-fold)↑ IDV brain-specific accumulation	[89] *^,^**
**Self-emulsifying drug delivery systems (SEDDS)**								
2013	**SMEDDS** **Lipid phase:** Capmul^®^ MCM (C8) (10%), Cremophor^®^ RH 40 (81%) **Aqueous phase:** PEG 300 (9%) **Fill SMEDDS into a hard gelatine capsule**	Micronized UC-781	N.D.	N.D.	+20.5 ± 0.52 to +32.0 ± 0.02	Smallest droplet sizes (14.9 ± 0.9,12.8 ± 0.4 and 16.1 ± 0.7) with the lowest oilcontent (1:9, 2:8 and 3:7 oil to surfactant/Cosurfactant ratios)The concentration of oil above 40% *w*/*w*: droplet size increased to as high as 100 nm	↓ solubility as the ↑ oil component (higher solubility of UC-781 in the surfactant and cosurfactant compared to oil)↑ in droplet size as ↑ oil UC781 had no significant effect on droplet size, polydispersity index, or zeta potentialFaster UC-781 release (100% by 60 min) from SMEDDS↑ absorption across the model membrane than UC781 powder	[130] **
2016	**SNEDDS** **Lipid phase:** Eucalyptus oil (12%), *Smix* (Cremophor^®^ EL and Brij^®^35, 1:1) (12–18%) **Aqueous phase:** Transcutol^®^ P (0–24%)	EFZ ^b^ logP = 4.6	N.D.	N.D.	N.D.	21.97 ± 1.3 to 113.9 ± 4.8	↓ mean globule size as ↑ surfactant (*Smix*) ↑ mean globule size as ↑ co-surfactant (Trancutol^®^) Faster EFZ release (>80% by 30 min) from SNEDDS compared to the free drug (18.3% by 30 min)↑ oral bioavailability (2.63-fold) than the free drug	[131] *^,^**
2016	**SNEOF** **Lipid phase:** Maisine^®^ 35-1 (0.7) **Aqueous phase:** Tween^®^80:Transcutol^®^ HP (1:0.6) **S-SNEOF** Aeroperl^®^ (absorbing agent) Compressed tablet (MCC)	LPV ^b^ logP = 5.94	99.45 ± 0.59	N.D.	N.D.	SNEOF: 53.16SNEOF tablets: 80	↑ LPV release (60% by 10 min) Lymphatic uptake of LPV from SNEOF↑ rate and extent of oral bioavailability than the free drug	[66] *^,^**

Notes: ^a^ intravenous injection; ^b^ oral administration; ^c^ transdermal administration; ^d^ intranasal administration; ^N.D.^ no data * in vivo studies performed; ** in vitro studies performed; *** ex vivo studies performed. Abbreviations: ARV, antiretroviral; AUC, area under the curve; AZT, zidovudine; Brij^®^-35, polyoxyethylene (23) lauryl ether; Capmul^®^ MCM, mono/diglyceride of caprylic acid; Capryol^®^ 90, propylene glycol monocaprylate; Chol, cholesterol; C_max_ maximum concentration; CNS, central nervous system; Cremophor^®^ EL, castor oil fatty acids, ethoxylated glycerol ester; Cremophor^®^ RH 40, polyoxyl 40 hydrogenated castor oil; D.L., drug loading; DRV, darunavir; EFV, efavirenz; EPC, egg phosphatidylcholine; Gelucire^®^ 44/14, lauroyl polyoxyl-32 glycerides; HG, hydrogel; IDV, indinavir; Labrasol^®^, PEG-8 capric/caprylic glyceride; LPV, lopinavir; Maisine^®^ 35-1, glyceryl monolinoleate; MCC, microcrystalline cellulose; ME, microemulsion; N.D., no data; NE, nanoemulsion; OA, oleic acid; Oleic Plurol^®^, polyglyceryl 6 dioleate; PEG, polyethylene glycol; o/w, oil-in-water; *Smix*, surfactant and cosurfactant mix; SMEDDS, self-microemulsifying drug delivery systems; SNEDDS, self-nanoemulsifying drug delivery systems; SNEOF, self-nanoemulsifying oily formulations; S-SNEOF, solid self-nanoemulsifying oily formulations; SQV, saquinavir; SQVM, saquinavir mesylate; Transcutol^®^ HP, diethyleneglycol monoethyl ether; Transcutol^®^ P, diethylene glycol monoethyl ether.

## Data Availability

Data sharing not applicable.

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
