# Peer review of "Lipid Nanocarriers for Anti-HIV Therapeutics: A Focus on Physicochemical Properties and Biotechnological Advances"

_pharmaceutics, 2021, doi:10.3390/pharmaceutics13081294_

Round 1

Reviewer 1 Report

An excellently written, very comprehensive review including very recent data on lipid-based delivery systems.  The review is particularly accurate in summarizing and describing the studies reported so far.   Different lipid entities for drug delivery are outlined and the most relevant administration routes for drug delivery are discussed in great detail. The focus on ARV agents and HIV therapy makes the review special rather than the physicochemical description of lipid carriers.

Unfortunately, the review comes to an abrupt end.  I definitively miss a conclusion, an outlook or perspective.  A personal view could also be elaborated at least in some detail.

I would also have liked to see one or two overview pictures on potential ARV drug administration routes and respective targets.

Reviewer 2 Report

The main issue with this manuscript is readability. In order to improve it, please give the reader the following itens: 1) One figure on molecular components of the virus and their organization plus virus-host interaction determining infection. 2)One Figure with chemical structure of antivirals. 3)One figure with major mechanisms of infection. 4)One section discussing examples of antivirals resistance (molecular basis and mechanisms leading to drugs resistance) 5)Clear description of two or three examples on how lipid carriers can imrpove delivery, abailability and targeting (improve references search on this topic) 6)Describe mechanism of si RNA activity and include a better account on Nobel Prize work (quote and describe seminal references). 7)Use at least two illustrative figures per section.

Reviewer 3 Report

Dear Authors,

Re: [pharmaceutics-1325133]

Title:  "Lipid nanocarriers for anti-HIV therapeutics: a focus on physicochemical properties and biotechnological advances"   I enjoyed reading your manuscript which was a sort of nostalgic scientific topic for me (took Virology course about 27 years ago and HIV, as a challenge pathogen back then, was my favourite). Please consider my below comments: 1. In the Abstract: please correct: "The current review focus";   2. In the Introduction: please change "associated to multi-regimen" to "associated with the ...";   3. At the end of the Introduction you mentioned: "To the best of our knowledge, no similar reviews have considered the composition and characterization of lipid nanocarriers in terms of size, colloidal stability, ...".  Just to let you know, In the last 3 years, we have published few articles, one in the same journal, a highly-cited article, exactly on the topic you mentioned: i)   https://doi.org/10.3390/pharmaceutics10020057 ii)  https://doi.org/10.1016/j.heliyon.2018.e01088 iii)   https://doi.org/10.3390/biomedicines9050520   4. In section 2: please specify U.S. FDA to distinguish it from FDA of other countries;   5. In section 2, Line 128: please add "and" between "non-nucleoside RT inhibitors (NNRTIs)" and "protease inhibitors";   6. Line 150: please correct the sentence: "while drug associations may improve";   7.  Line 152: please change "associated to toxic side effects" to "associated with";   8. In section 3: please add the statement "upon input of energy" after "self-organize" and "self-assembly". Self organisation of lipid / phospholipid ingredients of the nanocarriers only occurs after energy is provided to the system.;   9.  Please double check: "amphiphilic hydrocarbon building blocks" (this may narrow down and limit the choice of ingredients used to construct lipidic nanocarriers), and "hydrogen bonding" (Lines 191 - 193);   10. In section 3.1.: please split Citations for Gagne et al.  and  Sudhakar et al.;   11. Some abbreviations are not defined, I suggest adding a "List of Abbreviation" for the benefit of readers (in addition to the abbreviation list below Tables);   12. In line number 362 add: "Tocosome" to the list you mentioned: "new classes of lipid vesicles known as transferosomes, niosomes, ethosomes, and cubosomes ...";   13. In line 370, please change "than" to "then" (If systemic administration is intended than drugs should have hydrophobic properties);   14. Please rephrase the sentence: "An example of
(Line 409) mucopenetrating strategy is the coating with PEG used Pokharkar et al. and ...";
  15. Please correct: "longer-fasting" (should be: "longer-lasting";   16. The rationale for describing AD at the end of manuscript needs more explanations in the Introduction and where the paragraph is located;   17. Readers would highly benefit if you add a Conclusion / Synopsis section at the end of your article.       

Round 2

Reviewer 2 Report

The figures helped to catch the reader attention. 

I still think that molecular structures of ARV drugs and a better discussion on their interaction with the lipid carriers would further contribute to readability.

Author Response

According to the reviewer suggestion we have included 4 additional figures (the manuscript has now 8 figures in total) containing the molecular structures of ARV drugs. A better discussion on their interaction with the lipid carriers was also added, including examples of how lipid carriers can improve delivery, bioavailability and targeting (an additional section with targeting strategies was added and references improved on this topic.)